# PINK1 drives production of mtDNA-containing extracellular vesicles to promote invasiveness

Nicolas Rabas[1]*, Sarah Palmer[1]*, Louise Mitchell[1], Shehab Ismail[1], Andrea Gohlke[1], Joel S. Riley[1,2], Stephen W.G. Tait[1,2], Payam Gammage[1,2], Leandro Lemgruber Soares[3], Iain R. Macpherson[1,2], and Jim C. Norman[1,2]

The cystine-glutamate antiporter, xCT, supports a glutathione synthesis program enabling cancer cells to cope with metabolically stressful microenvironments. Up-regulated xCT, in combination with glutaminolysis, leads to increased extracellular glutamate, which promotes invasive behavior by activating metabotropic glutamate receptor 3 (mGluR3). Here we show that activation of mGluR3 in breast cancer cells activates Rab27-dependent release of extracellular vesicles (EVs), which can transfer invasive characteristics to "recipient" tumor cells. These EVs contain mitochondrial DNA (mtDNA), which is packaged via a PINK1-dependent mechanism. We highlight mtDNA as a key EV cargo necessary and sufficient for intercellular transfer of invasive behavior by activating Toll-like receptor 9 in recipient cells, and this involves increased endosomal trafficking of pro-invasive receptors. We propose that an EV-mediated mechanism, through which altered cellular metabolism in one cell influences endosomal trafficking in other cells, is key to generation and dissemination of pro-invasive microenvironments during mammary carcinoma progression.

## Introduction

Tumor cells up-regulate glutaminolysis to support anaplerosis and growth while establishing cytoprotective programs, such as glutathione synthesis, to counteract metabolically stressful tumor microenvironments (Zhu and Thompson, 2019). Key to glutathione synthesis is the xCT cystine-glutamate antiporter (SLC7A11), which imports cystine at the expense of equimolar glutamate efflux (Combs and DeNicola, 2019), and this leads to increased glutamate in the extracellular milieu. Increased circulating glutamate has been reported in patients with cancer, and in the MMTV-PyMT autochthonous mouse model of mammary carcinoma using mice expressing the polyoma-middle-T (PyMT) oncogene under the control of the mammary epithelial mouse mammary tumor virus (MMTV) promoter, serum glutamate levels correlate closely with tumor burden (Budczies et al., 2015; Dornier et al., 2017). Extracellular glutamate activates metabotropic glutamate receptor 3 (mGluR3), and this drives endosomal trafficking of the membrane type 1 matrix metalloproteinase (MT1-MMP) to promote disruption of basement membranes, which contributes to invasiveness, and lung colonization by mammary carcinoma cells is opposed by i.p. administration of a small-molecule inhibitor of mGluR3 (Dornier et al., 2017).

Mobilization of endosomal MT1-MMP requires Rab27-dependent delivery of late endosomes to the plasma membrane (Dornier et al., 2017; Macpherson et al., 2014), and fusion of late endosomes with the plasma membrane also leads to release of extracellular vesicles (EVs; Ostrowski et al., 2010). EVs are important mediators of intercellular communication during cancer progression and can render certain organs, such as the lung and liver, more susceptible to metastatic seeding, a process that has been termed "metastatic niche priming," due to recruitment of myeloid cells to, and on ECM deposition within, metastatic target organs (Wortzel et al., 2019). Thus, it is key to understand the mechanisms underlying how EVs mediate communication between cells and organs. A range of potential EV cargoes include oncoproteins, mRNAs, miRNAs, and DNA, and many of these may mediate transfer of various phenotypes between cancerous and other cells. However, before we can assemble a full picture of EV-mediated intercellular communication, it is necessary to understand the mechanistic details underlying the processes that enable this cascade, including identification of relevant transferrable EV cargoes and their sorting within donor cells, and determination of the cellular

[1]Beatson Institute for Cancer Research, Glasgow, UK;   [2]Institute of Cancer Sciences, University of Glasgow, Glasgow, UK;   [3]Glasgow Imaging Facility, Institute of Infection, Immunity and Inflammation, University of Glasgow, Glasgow, UK.

Correspondence to Jim C. Norman: j.norman@beatson.gla.ac.uk

*N. Rabas and S. Palmer share first authorship; N. Rabas's present address is Francis Crick Institute, London, UK;   Shehab Ismail's present address is Department of Chemistry, Catholic University of Leuven, Heverlee, Belgium.

signaling systems that allow recipient cells to respond to tumor EVs. Mechanisms underlying some of these processes have been elucidated. Mutant p53-expressing pancreatic tumors influence the ECM microenvironment of the lung by altering sorting of podocalyxin into EVs (Novo et al., 2018). In turn, podocalyxin-containing EVs influence ECM deposition by altering integrin recycling in fibroblasts, highlighting the endosomal system of recipient cells as a key element of the cellular response to EVs.

In the present study, we report that the combination of increased extracellular glutamate and altered mitochondrial function leads to increased production of EVs, which promote endosomal trafficking and invasive behavior in recipient cells. We show that the mitochondrial chromosome is packaged into these EVs via a mechanism that involves phosphatase and tensin homolog–induced putative kinase 1 (PINK1)-dependent interaction of late endosomes with mitochondria. Finally, by reducing mtDNA in exosomes through the production of Rho0 cells, and by using synthetic unilamellar liposomes to mimic mitochondrial DNA (mtDNA)-containing EVs, we demonstrate that this cargo is necessary and sufficient to evoke trafficking and invasive responses in recipient cells via activation of Toll-like receptor 9 (TLR9).

## Results

### mGluR3 drives Rab27/CD63-dependent EV release
Glutaminolysis drives invasive behavior by increasing extracellular glutamate, which activates mGluR3 to promote Rab27-dependent trafficking of MT1-MMP (Dornier et al., 2017), and Rab27s also control EV release (Ostrowski et al., 2010). Therefore, we used differential centrifugation followed by nanoparticle tracking analysis (NTA) and Western blotting to determine whether mGluR3 drives EV release. Treatment of MDA-MB-231 breast cancer cells with the mGluR3 antagonist LY341495 (LY95) reduced EV release (Fig. 1 a). Consistently, following glutamine starvation (to deplete extracellular glutamate), addition of glutamate to the extracellular medium increased release of EVs (Fig. 1 b). To further establish a role for mGluR3 in EV release, we used immunogold labeling for the EV marker CD63 in combination with transmission EM. This indicated that CD63-positive EVs released by MDA-MB-231 cells were ~100 nm in diameter and that mGluR3 inhibition decreased release of these (Fig. 1 c). EV release was also opposed by combined siRNA of Rab27a and Rab27b (Fig. S1 a). In addition to being a marker for EVs, CD63 plays an active role in intraluminal budding from multivesicular bodies (MVBs; van Niel et al., 2011). Consistently, siRNA of CD63 reduced EV release from MDA-MB-231 cells (Fig. S1 b).

### mGluR3 and EV release are required to maintain mitochondrial function
We considered whether mGluR3 might influence oxidative respiration. A 5-d exposure to LY95 significantly reduced basal and maximum oxygen consumption rate (OCR; Figs. 2 a and Fig. S2 a) without altering the mitochondrial mass or membrane potential (Fig. S2 b). A shorter exposure (18 h) to LY95 was ineffective in reducing OCR, indicating that defects in mitochondrial

respiration need several days to accrue following inhibition of mGluR3 (Fig. S2 c). To determine whether mGluR3's influence on oxidative respiration was mediated via control of endosomal function and/or EV production, we measured OCR following Rab27 or CD63 knockdown (Fig. S2 d). OCR, but not mitochondrial mass, was reduced following siRNA of Rab27 or CD63 for 5 d, with CD63 knockdown being most effective (Fig. 2 a; and Fig. S2, a and b). These data indicate that proper function of the late endosomal/MVB network and EV production is required to maintain mitochondrial function.

### Mitochondrial proteins and DNA are packaged into EVs
Mitochondrial proteins are commonly present in EV proteomes (Choi et al., 2015). Western blotting indicated that mitochondrial components such as glutamate dehydrogenase 1 (GLUD1; matrix), voltage-dependent anion channel 1 (VDAC; outer membrane), and cyclophilin D (matrix) are abundant in EV preparations. Consistently, addition of LY95 reduced the quantity of mitochondrial proteins associated with EV preparations (Fig. 2 b). Mitochondria possess 16-kbp circular genomes containing genes encoding respiratory chain subunits, mitochondria-specific tRNAs, and mitochondrial ribosomal RNAs. Quantitative PCR (qPCR) indicated that a range of mitochondrial genes were detectable in EV preparations (Fig. S3 a), and long-range PCR confirmed the presence of the intact mitochondrial chromosome (Fig. 2 c). mtDNA associated with EV preparations (including genes encoding tRNA-Leu, cytochrome c oxidase I [COX1], NADH dehydrogenase 1 [ND1], ND6, cytochrome b [CYB], and mtDNA nt positions 8339–9334) was more resistant to DNase than DNA encoding a nuclear gene (β-globin), indicating that the mitochondrial genome is preferentially packaged into the EV lumen (Fig. 2 d; Fig. S3 a). Consistently, inhibition of mGluR3 and knockdown of Rab27 or CD63 reduced release of mtDNA associated with EV preparations (Fig. 2 e and Fig. S3 b). Finally, we used sucrose density gradient centrifugation to confirm that mtDNA was associated with CD63-positive vesicles with a flotation density of 1.1–1.15 g/ml, consistent with that of EVs, and treatment with LY95 reduced this (Fig. 2 f).

### PINK1 controls production of mtDNA-containing EVs
PINK1 becomes stabilized at the surface of damaged/uncoupled mitochondria, where it phosphorylates and activates the E3-ubiquitin (E3-Ub) ligase Parkin and Ub moieties attached to mitochondrial proteins. This initiates phosphorylation/ubiquitylation cascades that promote mitophagy (Koyano et al., 2014; Lazarou et al., 2015; McWilliams and Muqit, 2017). Consistent with previous observations (Narendra et al., 2010), treatment with the mitochondrial uncoupler carbonyl cyanide 3-chlorophenylhydrazone (CCCP) elevated PINK1 levels and PINK1-dependent phospho-Ub levels (Fig. S4, a and b). We confirmed that prolonged suppression of PINK1 reduced basal and maximal OCR (Fig. S4 c). We then considered whether PINK1 might contribute to production of mtDNA-containing EVs. siRNA of PINK1 reduced release of mtDNA-containing EVs, as determined by differential centrifugation and sucrose density gradient centrifugation followed by NTA, Western blotting, and qPCR for the mitochondrial gene ND1 (Fig. 3, a and b). Furthermore, we confirmed that CRISPR depletion of PINK1

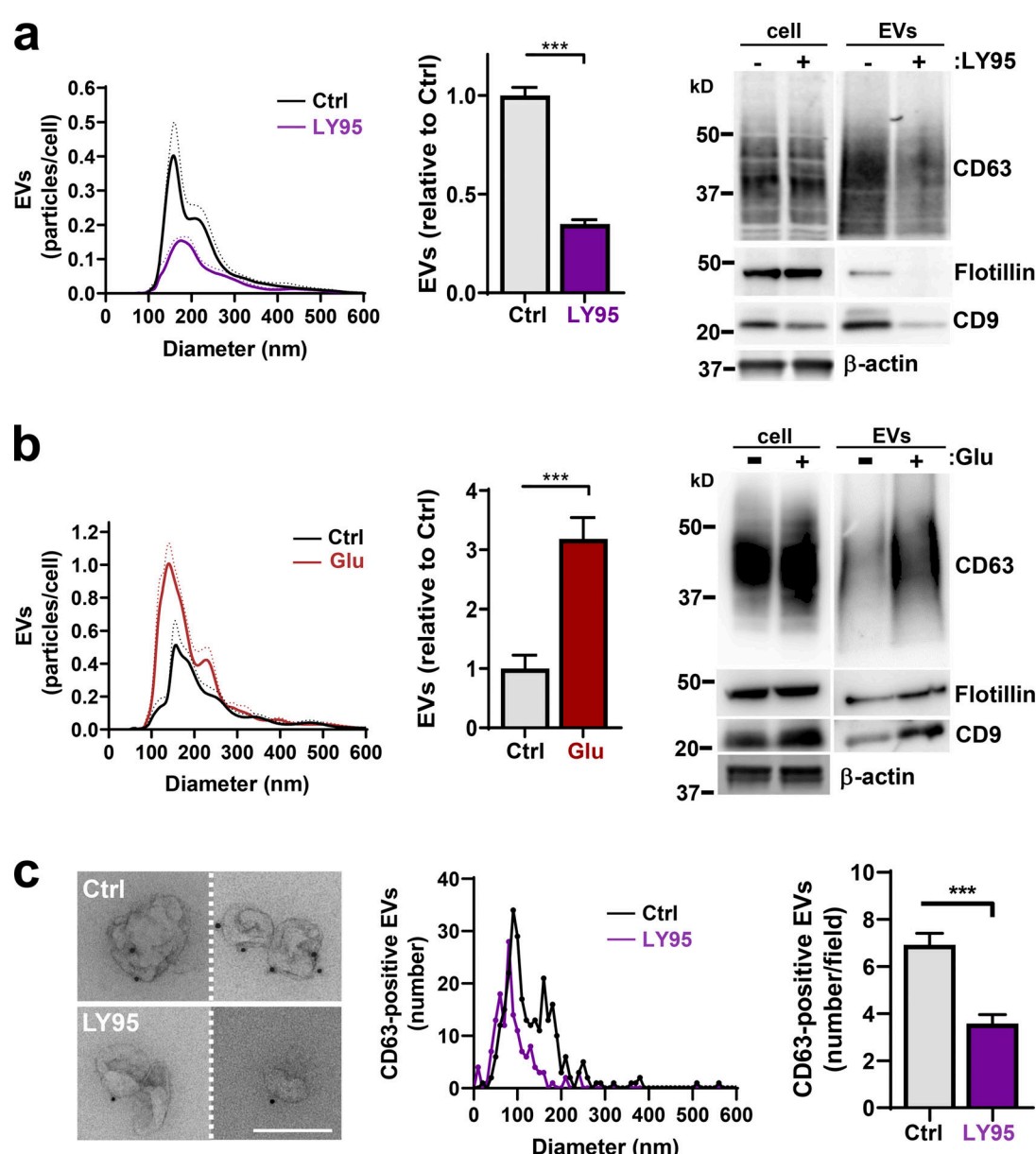

**Figure 1. Extracellular glutamate promotes EV release. (a and b)** MDA-MB-231 cells were incubated for 48 h in glutamine-replete medium with or without the mGluR3 antagonist LY95 (0.3 μM) or vehicle control (Ctrl; a) or in glutamine-deficient medium in the presence or absence of Glu (20 μM; b). EVs were isolated and analyzed by nanoparticle tracking. Values were normalized to cell number and are expressed as the number of particles of a given size in 0.5-nm increments. The bar graphs show the relative number of particles released per cell with diameter <160 nm, corresponding to the size of EVs. Values represent mean of $n = 4$ independent experiments. Cell lysates and differential centrifugation pellets were also analyzed by Western blotting for the indicated EV markers. Actin was used as a loading control. **(c)** Isolated EVs from cells treated with or without LY95 were fixed and placed onto formvar-carbon–coated grids. Dark spots indicate nanogold labeling of CD63. Bar, 200 nm. Diameter of EVs is plotted as the number of a given size in 10-nm increments, and total number of EVs per field is shown on the righthand graph from 40 different fields. Dotted lines and error bars represent SEM. ***, $P < 0.001$; all statistical analyses by $t$ test with Welch's correction.

opposed release of CD63-postive EVs from cells derived from a mouse mammary carcinoma driven by the PyMT oncogene (Fig. S4 d). Finally, we transfected PINK1 knockdown cells with a "rescue" vector encoding an siRNA-resistant YFP-tagged PINK1 (PINK1-YFP; Lazarou et al., 2015). Indeed, expression of PINK1-YFP rescued CCCP-driven levels of phospho-Ub and EV release following PINK1 knockdown (Fig. 3, c and d).

We investigated whether increased PINK1 levels might be sufficient to drive EV release. Addition of CCCP (to increase

PINK1 levels) increased EV release (Fig. 4 a), and this was opposed by siRNA of PINK1 or CD63 (Fig. 4 b). Overexpression of PINK1-YFP (Fig. 4 c) led to an increase in the release of EVs associated with mtDNA (Fig. 4, d and e). Moreover, a proportion of this EV-associated mtDNA is resistant to DNase and therefore is likely to be contained within these PINK1-driven EVs (Fig. 4 e).

We hypothesized that PINK1-dependent interaction between mitochondria and late endosomes/MVBs might underpin transfer of mitochondrial material to EVs. We used time-lapse

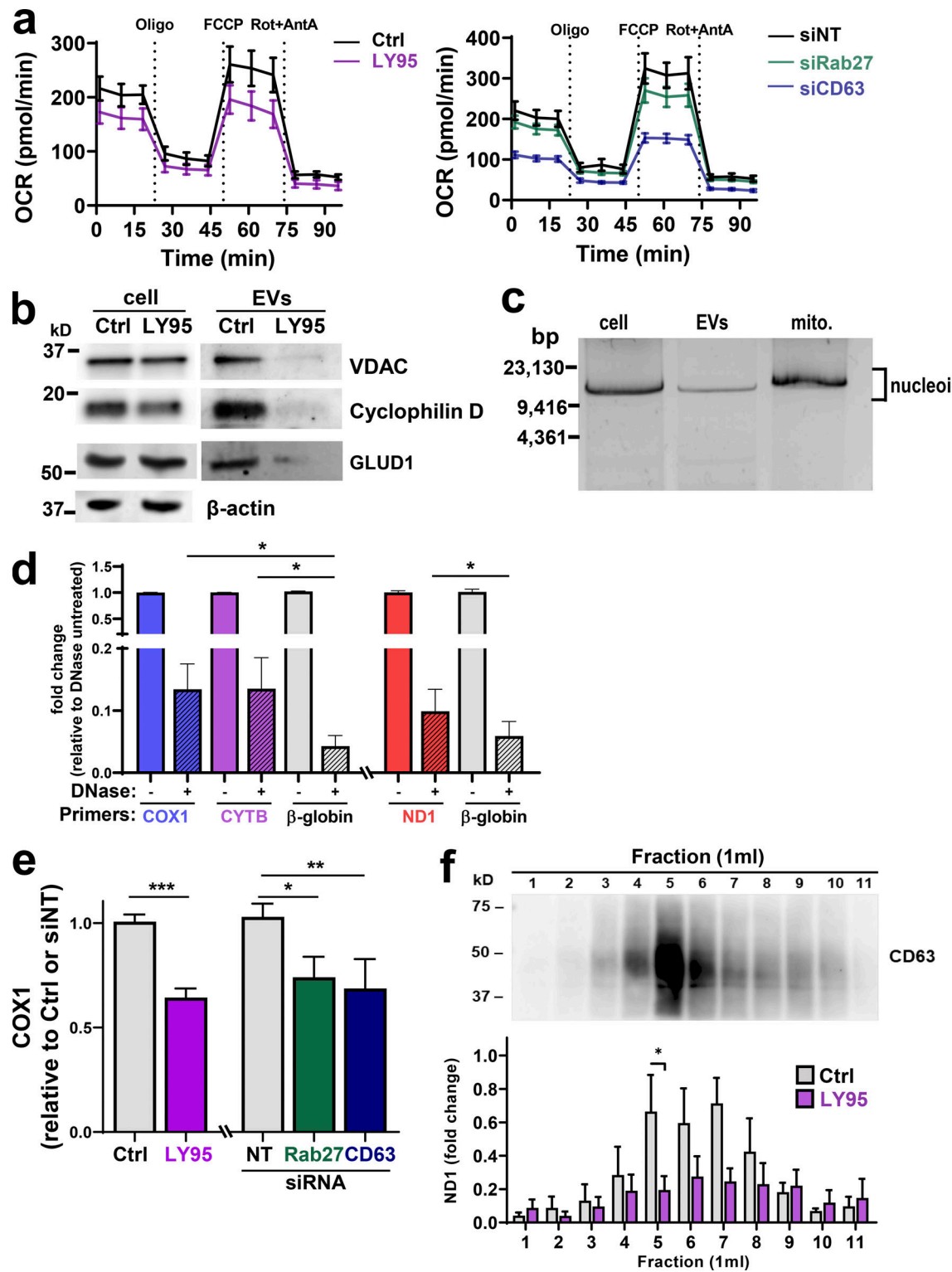

Figure 2. **Mitochondrial proteins and DNA are packaged into EVs. (a)** MDA-MB-231 cells were transfected with siRNAs targeting Rab27a/b (siRab27), CD63 (siCD63), or nontargeting siRNA (siNT) and incubated for 5 d in the presence (LY95) or absence (Ctrl) of LY95 (0.3 µM). Oxygen consumption was determined using the Seahorse XF[e]96 Extracellular Flux Analyzer. Sequential treatment with oligomycin, FCCP, and rotenone/antimycin A (Rot+AntA) was performed as indicated. Values represent the mean ± SEM of three independent experiments. **(b)** Cells were incubated in the presence of LY95 (0.3 µM) or vehicle control (Ctrl) for 48 h. EVs were isolated from media by differential centrifugation and analyzed by Western blotting for VDAC, cyclophilin D, and GLUD1. **(c)** Mitochondria (mito.) were purified from cells and EVs. mtDNA was extracted from these preparations and analyzed by long-range PCR. The expected migration position of the mitochondrial nucleoid (~16 × 10^6 D) is indicated. **(d)** EVs were purified by differential centrifugation and incubated in the presence and absence of DNase immobilized to agarose beads. mtDNA and nuclear DNA were determined by qPCR using primers complementary to sequences within the mitochondrial genes (COX1, CYTB, *n* = 5 independent experiments; or ND1, *n* = 3 independent experiments) or the nuclear β-globin genes. Values are mean ± SEM. *,

P < 0.05; paired *t* test. **(e)** Cells were transfected with siRNAs targeting CD63 (siCD63) or a nontargeting siRNA (siNT). Cells were then incubated for 48 h in the absence (Ctrl; NT and siCD63) or presence of 0.3 µM LY95. EVs were purified from the medium by differential centrifugation, and mtDNA content of these was determined using qPCR with primers recognizing the mitochondrial COX1 gene. Values are mean ± SEM from three independent experiments. *, P < 0.05; **, P < 0.002; ***, P < 0.001; Mann-Whitney *U* test. **(f)** Media collected in the absence (Ctrl) and presence of 0.3 µM LY95 were subjected to differential centrifugation. Differential centrifugation pellets were overlaid with a sucrose density gradient (2–0.4 M sucrose) and centrifuged at 200,000 *g* overnight. Gradients were eluted, and fractions were collected for analysis by qPCR for the mitochondrial ND1 gene and by Western blotting for CD63. Values represent mean ± SEM, *n* = 5 independent experiments. *, P < 0.05; Mann-Whitney *U* test.

fluorescence microscopy to determine the relationship between mitochondria (DsRed-mito) and late endosomes/MVBs (CD63-GFP). Mitochondria and late endosomes formed regular contacts that persisted for an average of ~70 s (Fig. 5 a; and Videos 1, 2, 3, and 4). The frequency of contacts was opposed by siRNA of PINK1 (Fig. 5 b). Moreover, the duration of contacts formed between DsRed-mito and CD63-GFP–positive structures was significantly increased by addition of CCCP and opposed by siRNA of PINK1 (Fig. 5 b). These data establish a role for PINK1 in promoting association between mitochondria and late endosomes/MVBs.

The autophagy machinery has recently been shown to regulate a novel secretory pathway, termed "microtubule-associated protein-1 light chain 3 (LC3)-dependent EV loading and secretion," which leads to release of EVs containing RNA-binding proteins (Leidal et al., 2020). Given PINK1's role in mitophagy, an autophagic process, we determined whether the autophagy regulators autophagy-related 5 (Atg5) and Atg7 and the mitophagy gatekeeper FAK family interacting protein of 200 kD (FIP200; Vargas et al., 2019) might contribute to EV production. However, neither siRNA of Atg5/7 (Fig. S5, a and b) nor suppression of FIP200 (Fig. S5, c and d) opposed release of EVs.

PINK1's contribution to mitophagy is via phosphorylation of Parkin and the Ubs conjugated to mitochondrial proteins, which then engage the autophagy machinery (Koyano et al., 2014; Lazarou et al., 2015; McWilliams and Muqit, 2017). Given our finding that release of EVs associated with mtDNA is not ATG5/7 dependent, we wished to determine whether there is a role for PINK's kinase activity in EV release. We transfected PINK1 knockdown cells with a PINK1-YFP rescue vector containing three point/substitution mutations (K219A, D362A, and D384A) to render the protein kinase dead (Beilina et al., 2005). Surprisingly, expression of this kinase-dead PINK1 completely restored EV release in PINK1 knockdown cells, despite the fact that this construct was unable to rescue CCCP-driven phospho-Ub levels following siRNA of PINK1 (Fig. 3, c and d). Taken together, these data indicate that PINK1 promotes physical interaction of late endosomes with mitochondria and contributes to release of mtDNA-containing EVs via a mechanism that is distinct from PINK's canonical, kinase-dependent role in mitophagy.

### mtDNA-containing EVs promote endosomal trafficking and invasiveness via TLR9

EVs can alter cell migration in recipient cells by promoting endosomal trafficking (Novo et al., 2018). We determined whether EVs released from "donor" cells might influence trafficking of MT1-MMP, a receptor linked to invasiveness, in "recipient" cells. We collected conditioned media from donor cells in the presence and absence of LY95, and we purified EVs from these. EVs were then incubated with mCherry-MT1-MMP–expressing recipient cells that had previously been starved of glutamine to deplete extracellular glutamate and suppress MT1-MMP trafficking (Fig. 6 a). Trafficking of mCherry-MT1-MMP vesicles to the ventral plasma membrane of recipient cells was then determined using total internal reflection fluorescence (TIRF) microscopy (Koyano et al., 2014; Lazarou et al., 2015; McWilliams and Muqit, 2017). EVs from glutamine-replete donor cells increased delivery of mCherry-MT1-MMP–positive vesicles to the plasma membrane of recipient cells, whereas EVs from LY95-treated donors were less effective in this regard (Fig. 6 b). Consistently, donor cell EVs enabled glutamine-starved recipient cells to execute MT1-MMP–dependent invasive processes, such as collagen degradation (Fig. 6 c) and invasion, into 3D fibroblast-derived ECM (Fig. 6 d). By contrast, EVs from donor cells in which mGluR3 was inhibited or PINK1 was knocked down were unable to drive collagen degradation or invasive behavior (Fig. 6, c and d). Consistently, suppression of Rab27 or CD63 in donor cells also rendered EV preparations to be ineffective in driving invasiveness in recipient cells (Fig. 6, c and d).

Although DNase protection approaches indicated that a higher proportion of EV-associated mtDNA than nuclear DNA is likely to be encapsulated within the EVs, we wished to determine the specificity of mtDNA in the ability of EVs to evoke invasive responses from recipient cells. We generated mtDNA-deficient "Rho0" cells and found that these release quantities of exosomes similar to those of control cells, but with reduced mtDNA (but not nuclear DNA) content (Fig. 7, a and b). Interestingly, exosomes from Rho0 cells are unable to promote collagen degradation (Fig. 7 c) or to drive extension of invasive protrusions (Fig. 7 d) when they are incubated with glutamine-starved recipient cells. These data indicate that it is specifically the mitochondrial (as opposed the nuclear) DNA associated with tumor EVs that is required for these EVs to evoke invasive responses.

The TLR family mediates responses to moieties associated with infective agents and cell damage, and certain TLRs are localized to endosomes (Luo et al., 2020). Because TLR9 is an endosomally localized receptor that is activated by double-stranded DNA, this is a good candidate for mediating cellular responses to the mtDNA associated with tumor cell EVs. Direct activation of TLR9 using a specific agonist (oligodeoxynucleotide [ODN] 2006) drove collagen degradation and extension of invasive protrusions in glutamine-starved cells (Fig. 8, a and b). Consistently, pharmacological blockade (using ODN-TTAGGG-A151) of TLR9 opposed the ability of mtDNA-containing EVs to drive extension of invasive protrusions (Fig. 8 c). Moreover, siRNA of TLR9 EVs inhibited invasive behaviors, including extension of protrusions (Fig. 8 c) and collagen degradation (Fig. 8 d), evoked by tumor cell EVs in glutamine-starved recipient cells. These data indicate that cancer cells (in which

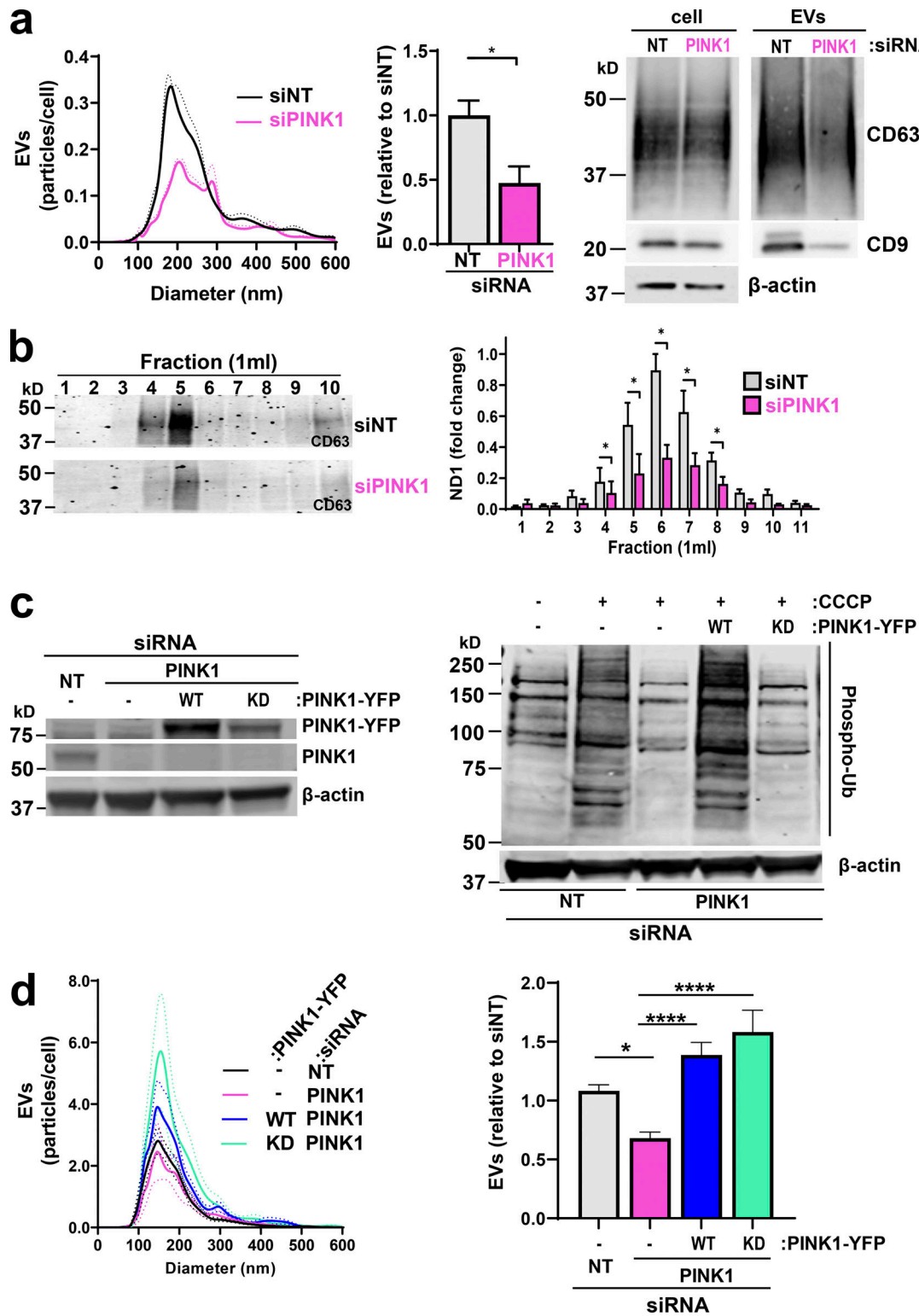

Figure 3. **PINK1 controls production of mtDNA-containing EVs. (a and b)** MDA-MB-231 cells were transfected with an siRNA targeting PINK1 (siPINK1) or a nontargeting control siRNA (siNT). EVs released from transfected cells were analyzed using nanoparticle tracking and Western blotting for CD63 or CD9 as in Fig. 1 a. Values represent the mean, and dotted line and error bars represent the SEM; n = 5 independent experiments. *, P < 0.05; t test with Welch's correction (a). Differential centrifugation pellets were further fractionated using sucrose density gradient centrifugation and analyzed by qPCR for mtDNA and by Western blotting for CD63 as in Fig. 2 f (b). Values represent mean ± SEM; n = 3 independent experiments. *, P < 0.05; Mann-Whitney U test. **(c and d)** Cells were transfected with an siRNA targeting PINK1 (siPINK1) or nontargeting control siRNA (siNT) in combination with rescue vectors for WT or kinase-dead (KD) PINK1-YFP. Cells were treated with CCCP as indicated, lysed, and analyzed by Western blotting for PINK1 or phosphorylated Ub with actin as a loading control (c). EVs were collected over a 48-h period and analyzed using nanoparticle tracking as in Fig. 1 a (d). Values represent the mean of four independent experiments; dotted line and error bars are SEM. *, P < 0.01; ****, P < 0.0001; one-way ANOVA with Dunn's multiple comparison test.

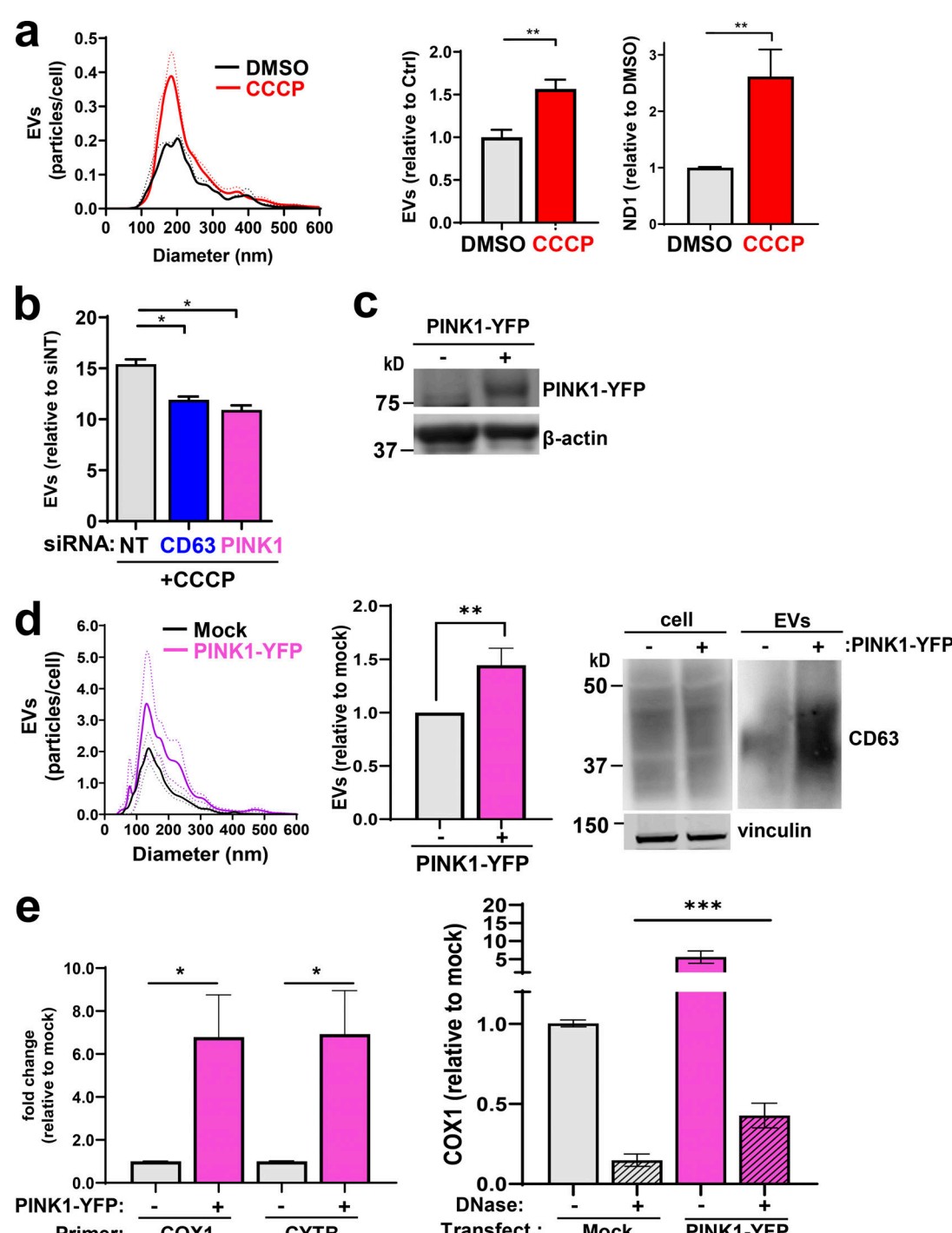

Figure 4. **Increased PINK1 levels are linked to EV release. (a)** MDA-MB-231 cells were treated with CCCP (10 µM) or control vehicle (DMSO), and released EVs were purified using differential centrifugation and analyzed as in Fig. 2 e. Values represent the mean of three independent experiments; dotted line and error bars represent SEM. **, P < 0.01; t test with Welch's correction. **(b)** Cells were transfected with siRNAs targeting CD63, PINK1, or a nontargeting control (siNT) and treated with CCCP (5 µM). EVs were then purified and analyzed by nanoparticle tracking. Values represent the mean ± SEM of three independent experiments. *, P < 0.05; t test with Welch's correction. **(c–e)** Cells were transfected with a vector encoding PINK1-YFP or empty vector (mock). EVs were collected from transfected cells and their number, size distribution, and CD63 and mtDNA (both DNase-sensitive and DNase-resistant) content were determined as in Figs. 1 a and 2 d. Values are the mean ± SEM; n = 4 technical replicates. *, P < 0.05; **, P < 0.001; Mann-Whitney U test. In the righthand graph of e, values are mean ± SEM, n = 5. ***, P < 0.001; Wilcoxon test.

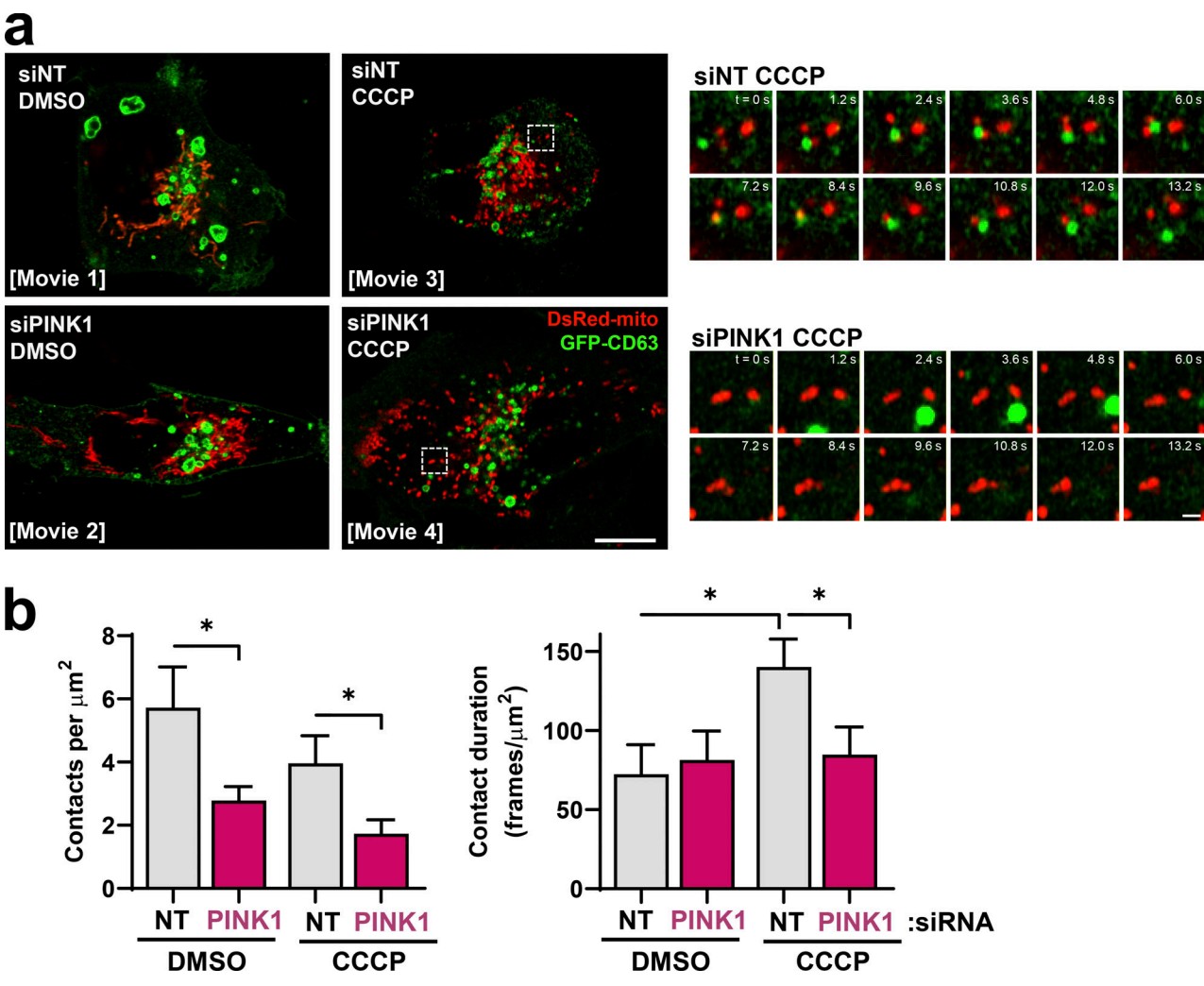

Figure 5. **PINK1 contacts with CD63-positive endosomes. (a and b)** MDA-MB-231 cells were transfected with GFP-tagged CD63 (GFP-CD63; green) and mitochondrial targeted DsRed (DsRed-mito; red) in combination with siRNAs targeting PINK1 (siPINK1) or nontargeting control (siNT). Transfected cells were incubated in the absence (DMSO) or presence (CCCP) of CCCP (10 μM) and imaged using a Zeiss LSM 880 Airyscan confocal microscope at 37°C with a frame rate of 0.59 s$^{-1}$. Videos 1, 2, 3, and 4 were assembled from these images, and the presented images depict representative stills from the videos. Bar, 10 μm; bar in zoom, 1 μm. The number and duration of contacts between DsRed-mito– and GFP-CD63–positive structures were determined and are plotted as mean ± SEM; *n* = 3 independent experiments. *, P < 0.05; Mann-Whitney *U* test.

glutaminolysis is active) release mtDNA-containing EVs that transfer endosomal trafficking and invasive characteristics to glutamine-starved cells via activation of TLR9.

### Membrane-encapsulated mtDNA is necessary and sufficient to promote trafficking and invasiveness

EVs contain several cargoes, including mtDNA, with potential to activate TLRs. We adopted a reconstitution approach to determine whether the mtDNA was itself sufficient to drive endosomal trafficking and invasive behavior in recipient cells. We constructed unilamellar liposomes with dimensions similar to those of EVs (90–120-nm diameter), loaded them with mtDNA, and treated them with immobilized DNase to remove DNA that was not lipid encapsulated (Fig. 9, a and b). Using capture ELISA recycling assays and TIRF microscopy, we determined that mtDNA-loaded liposomes promoted recycling of pro-invasive receptors, including MT1-MMP and α5β1 integrin, from endosomes

to the plasma membrane, whereas control liposomes were ineffective in this regard (Fig. 9 c). Moreover, EV-induced delivery of MT1-MMP–containing vesicles to the ventral plasma membrane was opposed by pharmacological inhibition of TLR9 (Fig. 9 d). Consistently, mtDNA-loaded liposomes restored invasive behaviors to glutamine-starved cells, and this was opposed by inhibition or siRNA of TLR9 (Fig. 9 e).

These data indicate that mtDNA is packaged via a PINK1-dependent mechanism into EVs released by breast cancer cells and that mtDNA is a key EV cargo both necessary and sufficient to transfer glutaminolysis-driven invasive phenotypes between cells.

### Discussion

We have identified a means of intercellular communication in which processes evoked to mitigate cytotoxicity in metabolically stressed cells can evoke invasive behavior in other cells. This

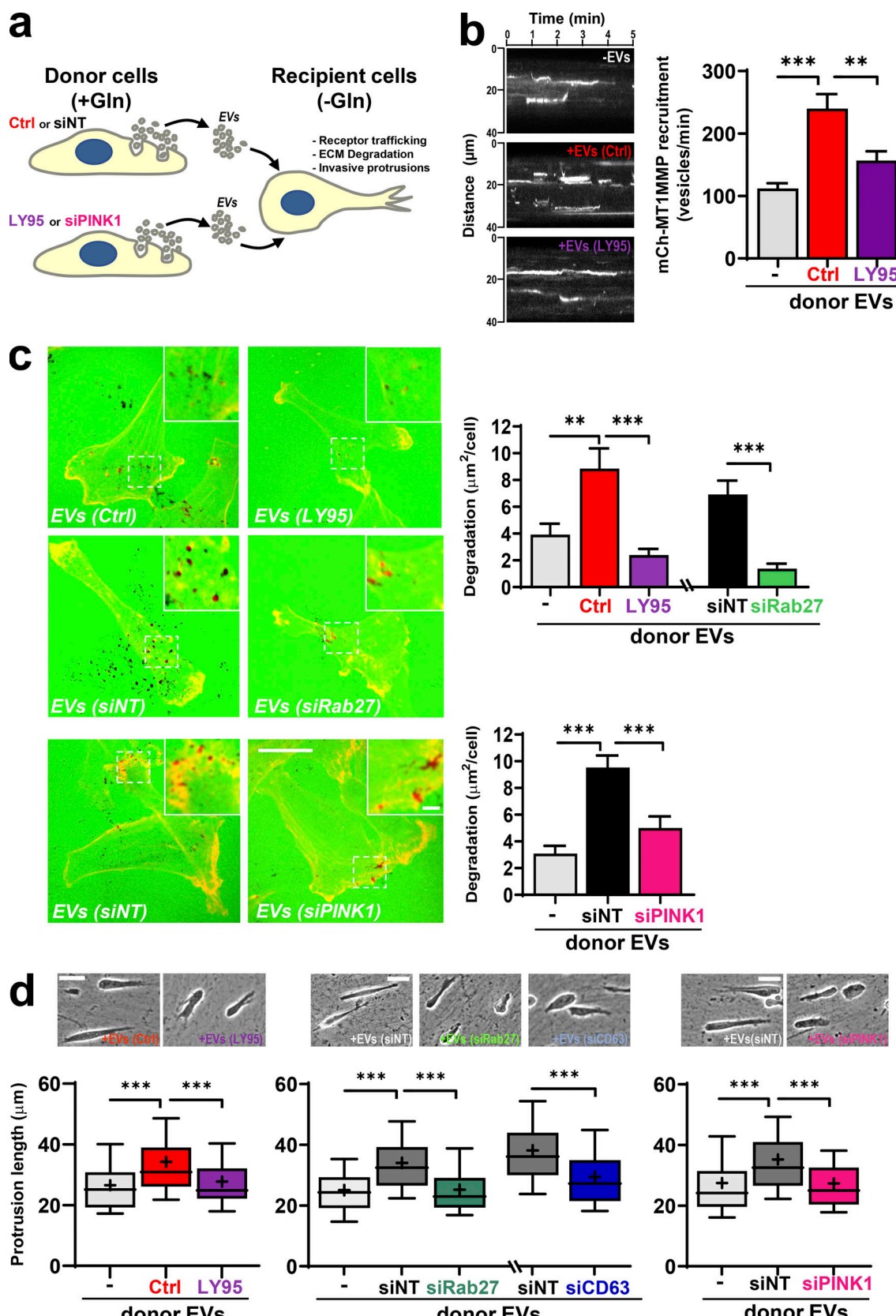

Figure 6. **mtDNA-containing EVs promote endosomal trafficking and invasiveness. (a)** Protocol to study the influence of EVs collected from glutamine-replete (+Gln) donor cells on the invasive behavior of Gln-starved (−Glu) recipient cells. **(b)** Donor MDA-MB-231 cells were incubated in glutamine-containing medium in the absence (Ctrl) and presence of LY95 0.3 µM for 48 h, and EVs were purified from this using differential centrifugation. Glutamine-starved recipient cells were incubated in the absence (−EVs) or presence of EVs from the indicated donor cells (+EVs [Ctrl]; +EVs [LY95]) for 3 d and then transfected

with mCherry-MT1-MMP and imaged using TIRF microscopy. Time-lapse videos were recorded with frames being collected every 1.5 s for 5 min. Kymographs of representative videos are displayed. The rate of appearance of mCherry-positive structures in the TIRF field was calculated using the ImageJ plugin TrackMate. Values represent mean ± SEM from three independent experiments. **, P < 0.002; ***, P < 0.001; two-way ANOVA with Dunn's multiple comparison test. **(c)** Glutamine-starved recipient cells were treated for 3 d with EVs from LY95- or vehicle-treated (Ctrl) donor cells (as for b) or donor cells that had been transfected with siRNAs targeting PINK1 (siPINK1), Rab27a/b (siRab27), or nontargeting control (siNT). Recipient cells were then replated onto dishes coated with fluorescently conjugated gelatin (FITC-gelatin) in the absence of glutamine, and imaging was performed using confocal microscopy. Bar, 20 μm; bar in zoom, 2 μm. Gelatin degradation was determined using ImageJ. Values are mean ± SEM; n = 2 independent experiments, n = 4 technical repeats with a minimum of 45 cells being quantified for each condition. **, P < 0.002; ***, P < 0.001; two-way ANOVA with Dunn's multiple comparison test. **(d)** Glutamine-starved recipient cells were pretreated for 3 d with EVs from LY95- or vehicle-treated (Ctrl) donor cells (as in b) or donor cells that had been transfected with siRNAs targeting PINK1 (siPINK1), Rab27a/b (siRab27), CD63 (siCD63), or nontargeting control (siNT). Recipient cells were plated into fibroblast-derived ECM and imaged using time-lapse video microscopy. The length of protrusions extending in the direction of migration (distance from the nucleus to cell front) was determined using ImageJ. Bar, 30 μm. Whiskers are 10th–90th percentiles; + represents the mean. ***, P < 0.001; two-way ANOVA with Dunn's multiple comparison test.

proceeds via three sequential mechanistic events. First, PINK1, activated in response to mitochondrial damage, promotes physical interaction of late endosomes with mitochondria. Second, increased secretion of glutamate activates mGluR3 and drives Rab27-dependent exocytosis of EVs loaded with mtDNA. Third, mtDNA transported within EVs activates a TLR9-dependent mechanism to promote pro-invasive endosomal trafficking in other cells (Fig. 10).

Communication of metabolic information to nearby cells may be conveyed in several ways. Metabolites can be exchanged between cells in proximity. Glycolytic and glutaminolytic tumor cells generate lactate and glutamate, which are exported by monocarboxylate transporters and the xCT antiporter, and these can be assimilated by neighboring cells (Doherty and Cleveland, 2013; Sonnewald and Schousboe, 2016; Whitaker-Menezes et al.,

2011). Extracellular metabolites may influence other cells by interacting with surface receptors, such as GPR81 lactate receptor, GPR91 for succinate, and mGluRs for glutamate (Ristic et al., 2017). Mice bearing invasive MMTV-PyMT–driven carcinoma display elevated serum glutamate (Dornier et al., 2017), indicating that tumor metabolites are not locally restricted, introducing the possibility that tumor metabolites can evoke systemic responses. Glutamine metabolism can impact the release of EVs with pro-tumorigenic properties (Fan et al., 2020), and EVs have a well-established role in mediating communication between tumors and metastatic target organs (Costa-Silva et al., 2015; Novo et al., 2018; Peinado et al., 2012). Our findings highlight the likelihood that tumor cells experiencing metabolic stress use EVs to communicate their metabolic state to other cells within the tumor and farther afield.

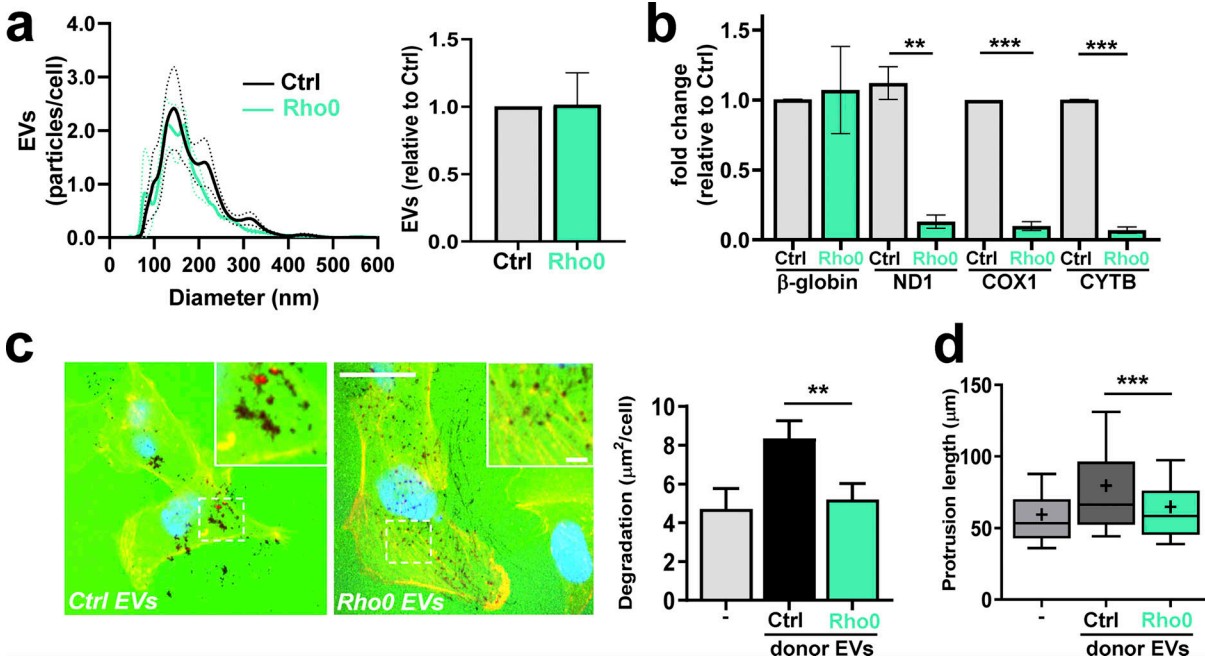

Figure 7.   **mtDNA is required for EVs to evoke invasive responses. (a and b)** Cells with reduced mtDNA content (Rho0) were generated by incubation of MDA-MB-231 cells with ethidium bromide for 5 d. EVs were collected from control (Ctrl.) and Rho0 cells over a 48-h period and analyzed to determine their number and size distribution (a) and nuclear and mtDNA content (b) as in Figs. 1 a and 2 d, respectively. **(c and d)** EVs from control (Crtl.) and Rho0 cells were then incubated with glutamine-starved cells, and the ability of these recipient cells to degrade collagen (c) and to extend invasive protrusions (d) was determined as in Fig. 6, c and d, respectively. Bar, 20 μm; bar in zoom, 2 μm. Values are mean ± SEM; n = 3 independent experiments. **, P < 0.002; ***, P < 0.001; ANOVA (one-way ANOVA for c; two-way ANOVA for d) with Dunn's multiple comparison test.

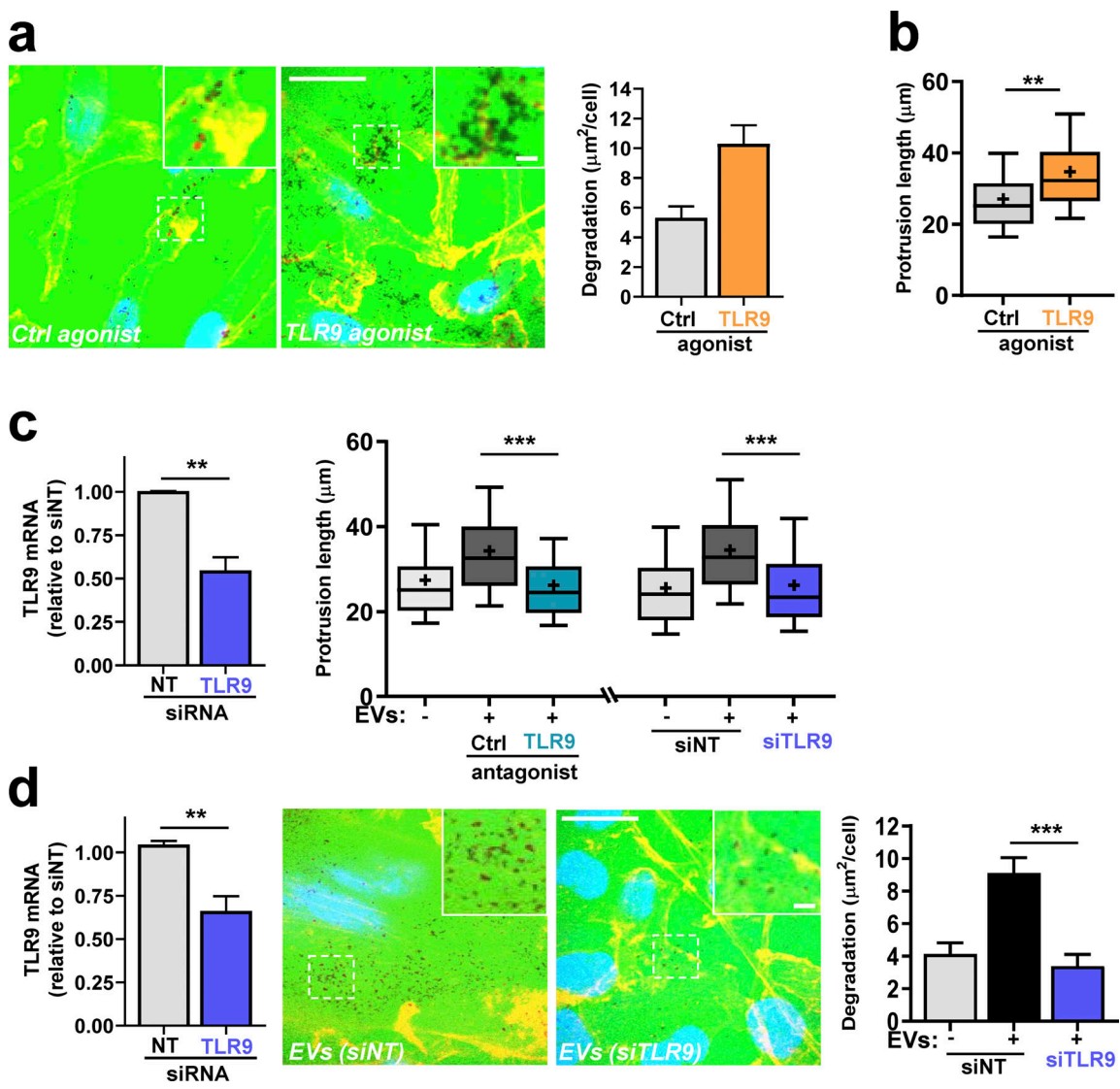

Figure 8. **EVs promote endosomal trafficking and invasiveness via a TLR9-dependent mechanism. (a and b)** Glutamine-starved cells were plated onto dishes coated with fluorescently conjugated gelatin (FITC-gelatin; a) or fibroblast-derived ECM (b) in the presence of ODN 2006 (TLR9 agonist; 1 µM) or ODN 2137 (control [Ctrl] agonist; 1 µM). 16 h following this, gelatin degradation (a) and invasive protrusion length (b) were determined as in Fig. 6, c and d, respectively. Values are mean ± SEM; n = 3 independent experiments. ***, P < 0.001; **, P < 0.002; Mann-Whitney *U* test. Bar, 20 µm; bar in zoom, 2 µm. **(c)** Recipient cells were transfected with siRNAs targeting TLR9 (siTLR9) or nontargeting siRNA (siNT). TLR9 knockdown was assessed using qPCR (left panel). Recipient cells were glutamine starved and incubated for 3 d in the absence (−) or presence (+) of EVs from glutamine-replete donor cells. Invasive protrusion length was then determined in the absence or presence of ODN-TTAGGG-A151 (TLR9 antagonist; 1 µM) or ODN-TTAGGG-control (Ctrl antagonist; 1 µM) as in Fig. 6 d. Whiskers are 10th–90th percentiles, + represents the mean. **, P < 0.01; ***, P < 0.001; two-way ANOVA with Dunn's multiple comparison test. **(d)** Recipient cells were transfected with stealth siRNAs targeting TLR9 (siTLR9) or nontargeting siRNA (siNT). TLR9 knockdown was assessed using qPCR (left panel). Transfected recipient cells were glutamine-starved, plated onto fluorescently conjugated gelatin, and incubated for 3 d in the absence (−) or presence (+) of EVs from glutamine-replete donor cells. Bar, 20 µm; bar in zoom, 2 µm. The ability of these recipient cells to degrade collagen was then determined as in Fig. 6 d. Values are mean ± SEM; n = 3 independent experiments. **, P < 0.01; ***, P < 0.001; one-way ANOVA with Dunn's multiple comparison test.

Donor cells influence recipient cell signaling and sorting of protein and nucleic acid cargoes into EVs. Proto-oncogenes and oncogenic receptors, such as EGF receptor and c-MET (Fan et al., 2020; Peinado et al., 2012; Zhang et al., 2017), can evoke transformed phenotypes in recipient cells, but how these receptors attain the appropriate membrane localization and topology in recipient cells is unclear. Annexin-6 and podocalyxin are EV cargoes known to promote metastatic niche priming. Annexin-6 promotes CCL2 induction in neutrophils (Keklikoglou et al.,

2019), but details of how this cargo is sorted into EVs and then interfaces with recipient cell signaling machinery are not known. Mechanistically better defined is podocalyxin's sorting into cancer cell EVs, which is controlled by Rab35, and this influences ECM deposition in metastatic target organs by influencing Rab-coupling protein–dependent integrin trafficking machinery in fibroblasts (Novo et al., 2018). miRNAs may alter expression of their cognate targets in recipient cells (Cooks et al., 2018), although to do this, they must access the mRNA

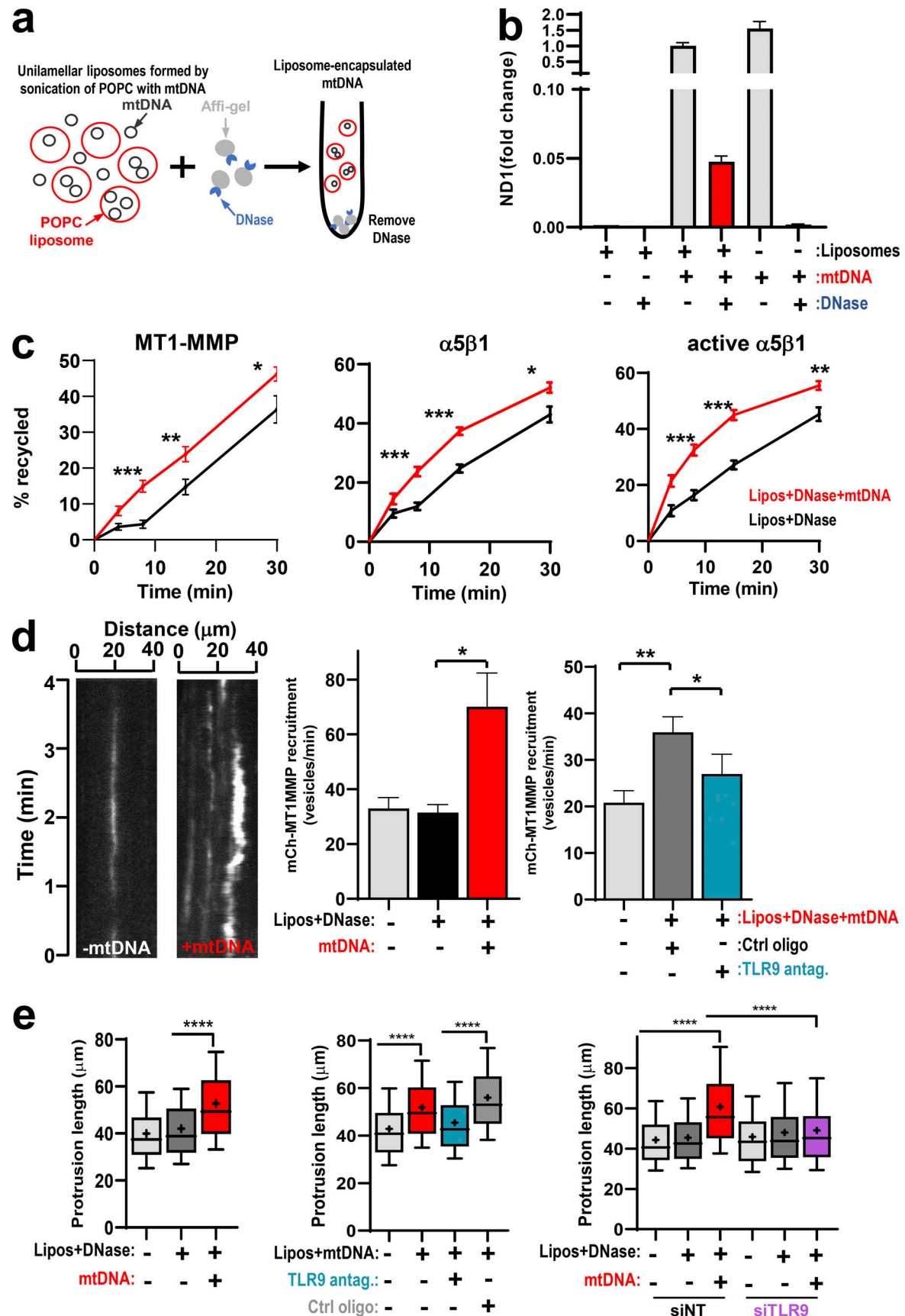

Figure 9.  **Membrane-encapsulated mtDNA promotes increased trafficking and invasiveness in a TLR9-dependent mechanism. (a)** Protocol for encapsulation of mtDNA within POPC liposomes, followed by removal of surface-associated mtDNA by treatment with DNase immobilized to Affi-Gel 10 agarose

beads. **(b)** POPC liposomes were formed in the absence and presence of mtDNA and treated with immobilized DNase as indicated. The mtDNA content of these was then determined using qPCR with primers recognizing the mitochondrial ND1 gene. Values represent mean ± SEM. **(c)** Glutamine-starved cells were incubated overnight with DNase-treated liposomes (lipos) that had previously been formed in the absence or presence (+mtDNA) of mtDNA. Recycling of MT1-MMP, $\alpha_5\beta_1$ integrin, and active $\alpha_5\beta_1$ integrin was determined using cell surface biotinylation/capture ELISA approaches. The proportion of receptors recycled to the plasma membrane is expressed as a percentage of the pool of the receptor labeled during the internalization period. Values are mean ± SEM; $n$ = 4 independent experiments. ***, $P < 0.001$; **, $P < 0.01$; *, $P < 0.05$; unpaired $t$ test. **(d)** Glutamine-starved cells were incubated for 3 d with DNase-treated liposomes (lipos) formed in the absence or presence of mtDNA. ODN-TTAGGG-A151 (TLR9 antagonist; 1 µM) or ODN-TTAGGG-control (Ctrl antagonist; 1 µM) was also included as indicated. Cells were then transfected with mCherry-MT1-MMP and imaged using TIRF microscopy as in Fig. 6 b. Values represent mean ± SEM from three independent experiments. *, $P < 0.05$; **, $P < 0.01$; two-way ANOVA with Dunn's multiple comparison test. **(e)** Glutamine-starved cells were treated with mtDNA-containing liposomes in the presence and absence of TLR9 antagonist as in d and transfected with siRNAs targeting TLR9 (siTLR9) or nontargeting siRNA (siNT). Cells were then plated into fibroblast-derived ECM, and the extension of invasive protrusions at the cell front was determined as in Fig. 6 d. Representative stills are presented. Bar, 30 µm. Whiskers are the 10th–90th percentiles; + represents the mean; $n$ = 3. ****, $P < 0.0001$; two-way ANOVA with Dunn's multiple comparison test.

processing and/or translational machinery in the cytosol by as yet undefined mechanisms. Extracellular nucleic acids evoke cellular responses by engaging with pattern recognition receptors, and miRNAs associated with tumor EVs have been shown to activate pattern recognition receptors. EVs from pancreatic cancer cells regulate TLR4 in dendritic cells via miR-203 (Zhou et al., 2014). However, although a recent report has identified the LC3-conjugation machinery as key to sorting a cohort of RNA-binding proteins and small noncoding RNAs into EVs (Leidal et al., 2020), metabolic landscapes that might influence packaging of nucleic acid cargoes into EVs are yet to be determined. Macrophages and dendritic cells express high levels of TLRs, indicating the likelihood that PINK1-dependent EV release offers a mechanism by which tumor cells undergoing

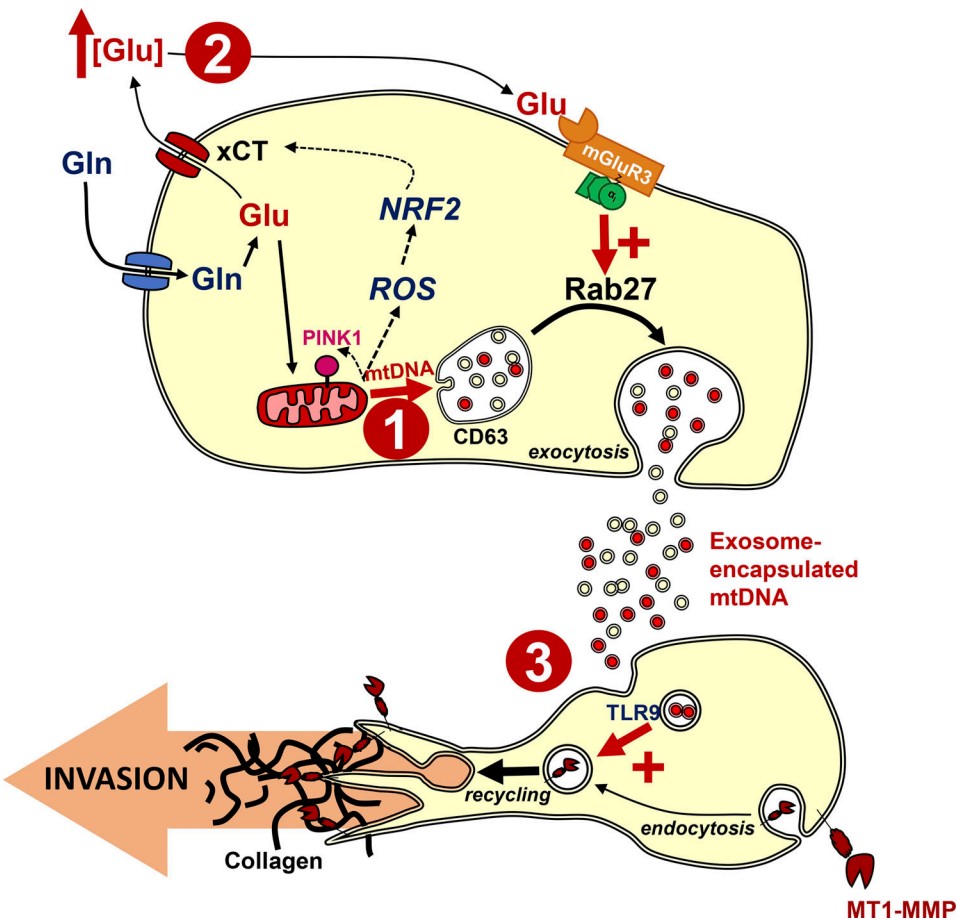

Figure 10. **Processes enlisted to mitigate cytotoxicity in metabolically stressed cells evoke invasive behavior in other cells.** Three sequential events contribute to mechanisms through which metabolic stress may be communicated between cells: (1) Mitochondrial damage/depolarization increases levels of PINK1 to promote physical interaction of late endosomes with mitochondria. This leads to transfer of the mitochondrial chromosome into the lumen of intraluminal vesicles of late endosomes; (2) a combination of glutaminolysis and up-regulation of xCT (SLC7A11) leads to increased secretion of glutamate to drive Rab27-dependent exocytosis of EVs loaded with mtDNA; and (3) mtDNA transported within these EVs activates a TLR9-dependent mechanism to promote pro-invasive endosomal trafficking of MT1-MMP in other cells.

mitochondrial stress may communicate with the innate immune system, and this will be an interesting line of investigation in the future.

When DNA is present in the cytosol, resulting from viral infection or following its (aberrant) release from the nucleus or mitochondria, this leads to immune and inflammatory responses by activating cytosolic DNA-sensing pathways such as cyclic GMP-AMP synthase (cGAS)/stimulator of interferon genes (STING; Riley et al., 2018) and AIM2 (absent in melanoma 2; Dang et al., 2017). Recently, activated T cells were shown to release mtDNA-containing EVs to stimulate cGAS/STING in dendritic cells (Torralba et al., 2018), but how EV-associated mtDNA can access the cytoplasmically located cGAS/STING machinery still presents a topological puzzle. Interestingly, these investigators reported that EV-associated mtDNA was more oxidized than endogenous mtDNA, suggesting that EV release from T cells may be linked to the altered redox metabolism. Cancer cells have also been found to release EVs in response to stresses associated with altered redox metabolism. Thermal and oxidative stress induces EV release from cancer cells (Hedlund et al., 2011; Wang et al., 2014). Moreover, EVs are released from breast tumors in response to treatment with taxanes and anthracyclines, forming a mechanism through which chemotherapeutics promote metastatic niche priming (Keklikoglou et al., 2019). However, although these investigators demonstrated a requirement for Rab27 in this process, mechanisms linking chemotherapy-induced metabolic stresses with EV release have not been addressed. Our data provide a mechanism for how a nucleic acid cargo (mtDNA) is packaged into EVs, and we provide insight into the metabolic landscapes that promote this. In the future, it will be interesting to investigate the role played by xCT-driven glutamate fluxes and PINK1 in EV release following chemotherapy and to determine whether the packaging of mtDNA into EVs influences metastatic niche priming following chemotherapy.

Mitochondria have been shown to be released from cells. Larger vesicles containing damaged mitochondria are released by *Caenorhabditis elegans* neurons and murine cardiomyocytes (Melentijevic et al., 2017; Nicolás-Ávila et al., 2020). More recently, neutrophils were reported to leave damaged mitochondria behind them as they migrate, and this has been suggested to constitute an alternative to mitophagy for maintenance of mitochondrial homeostasis and quality control (Jiao et al., 2021). However, it is likely that cells export mitochondria from cells for reasons other than to maintain mitochondrial homeostasis. Intercellular transfer of functional mitochondria has been reported to increase aerobic respiration in recipient cells (reviewed in Liu et al., 2021). EVs produced by stromal cells may transfer functionally intact mtDNA to cancer cells with impaired metabolism, allowing restoration of metabolic activity and release from metabolic quiescence (Sansone et al., 2017). Although we agree with these investigators that the intact mitochondrial genome is packaged into EVs, we have been unable to demonstrate any EV-mediated intercellular transfer of mitochondrial function. Indeed, we have purified EVs from cells with WT mitochondria and incubated these with recipient cells harboring a loss-of-function mutation in the mitochondrial ATP

synthase gene. In these experiments, EVs from cells with WT mitochondria did not rescue metabolic defects of cells with mutant mitochondria. Thus, we conclude that a primary function of PINK1-dependent sorting of mtDNA into EVs is to influence pro-invasive endosomal trafficking in recipient cells (via activation of TLR9), thus engaging in intercellular communication without direct transfer of functional mitochondrial components or genetic information.

PINK1 has an established role in mitochondrial homeostasis, and this is achieved via its ability to phosphorylate Parkin and Ub to increase phospho-Ub levels at the mitochondrial surface (Koyano et al., 2014; Lazarou et al., 2015; McWilliams and Muqit, 2017). This triggers recruitment of the ULK1 complex via the Ub-binding adaptor, FIP200 (Vargas et al., 2019). ATG proteins then mediate assembly of the autophagosome to encapsulate the damaged mitochondrion (Koyano et al., 2014; Lazarou et al., 2015; McWilliams and Muqit, 2017). Involvement of PINK1 in packaging of mtDNA into EVs suggests that mitophagy and mtDNA-containing EV production may share features. However, PINK1's kinase activity and the mitophagy machinery (ATG5/7 and FIP200) are not required for release of mtDNA-containing EVs. Consistently, a recent study has shown that mitochondrial material and mtDNA is routinely packaged into released EVs only when Parkin expression is absent (Todkar et al., 2021). Taken together, these observations indicate that PINK1 contributes to production of mtDNA-containing EVs in a way that is mechanistically distinct from its role in mitophagy. Also, although the two processes both involve congress between mitochondria and late endosomes/lysosomes, they do not share the same membrane topology. More topologically aligned to packaging of mtDNA into EVs is production of mitochondria-derived vesicles (MDVs). Vesicles are established to bud from mitochondria in a PINK1- and Parkin-dependent, but ATG5/7-independent, manner and these are trafficked to various intracellular locations, including the peroxisomes and late endosomes/lysosomes (Neuspiel et al., 2008; Sugiura et al., 2014; Yang and Yang, 2013). These connections to oxidative and degradative organelles have prompted proposals that MDV production may be an early response to oxidative stress that assists with reconditioning of damaged mitochondria in a manner more selective than mitophagy (McLelland et al., 2014; Soubannier et al., 2012). Our data clearly indicate that knockdown of Rab27 and CD63, factors with established roles in the production, plasma membrane delivery, and release of EVs, leads to attrition of mitochondrial function. Therefore, the likelihood that PINK1, acting in a kinase-independent manner, in conjunction with the endosomal regulators, CD63 and Rab27, to promote packaging of mitochondrial components into EVs may function in piecemeal or bit-by-bit mitophagy (as has been proposed for MDV production) must be considered, and it will be interesting to determine how PINK1 achieves this by acting as an adaptor rather than as a kinase.

Therapeutic responses and relapse of tumors with known oncogenic mutations, such as those in KRAS and p53, may be monitored using PCR detection of circulating DNA (Kilgour et al., 2020). Furthermore, circulating DNA increases following metabolic insults, such as paracetamol-induced hepatic injury, and this may be used to monitor these conditions (Laurent

et al., 2020). However, the use of circulating DNA as an early detection marker in cancer is limited because many of the mutations occurring in tumors are present in normal epithelia (Martincorena et al., 2018) and are not necessarily good markers of malignant or dysplastic disease. Our data indicate that adaptations enlisted by tumors to counter redox stress may be reflected in altered circulating factors. For instance, the glutamate release that results from up-regulation of the xCT transporter leads to detectably increased circulating glutamate in the serum in tumor-bearing subjects (Dornier et al., 2017). The present study provides a mechanistic rationale connecting increased glutamate secretion to release of mtDNA-containing EVs and demonstrates how these events are linked to invasive characteristics. Moreover, a recent study indicating that a number of tumor types harbor particular mutations in mtDNA may assist in identification of mtDNA-containing EVs in the circulation that have emanated from invasive cancer cells (Gorelick et al., 2021). Therefore, it will be interesting to determine the potential for combining measurement of circulating glutamate levels and EV-encapsulated mtDNA into a biomarker strategy for detection of breast cancer and for distinguishing indolent from more aggressive disease.

## Materials and methods

### Cell lines, antibodies, expression vectors, and reagents
MDA-MB-231 cells were maintained in DMEM (Life Technologies) supplemented with 10% FBS, 2 mM glutamine (unless otherwise stated), and antibiotics. A primary mouse breast cancer cell line was isolated from mammary tumors of mice expressing the PyMT oncogene under the control of the mammary epithelial MMTV promoter. Tumors from mice expressing the MMTV-PyMT transgene were surgically excised and minced using a McIlwain tissue chopper. Minced tumor specimens were transferred to DMEM supplemented with 10% FBS, 20 ng/ml EGF, 10 µg/ml insulin, 2 mM glutamine, and antibiotics and kept in culture for up to 20 passages.

The following antibodies were used for Western blotting and diluted 1:1,000 unless otherwise stated: hCD63 (BD Biosciences; 556019), mCD63 (Abcam; ab217345), CD9 (Cell Signaling Technology; 13174S), flotillin (Cell Signaling Technology; 18634), β-actin (Sigma-Aldrich; 1:5,000, 1978), Rab27A (Abcam; 55667), VDAC (Abcam; ab14734), cyclophilin D (Abcam; ab110324), GLUD1 (Abcam; ab153973), PINK1 (Novus Biologicals; BC100-494), LC3B (Cell Signaling Technology; 2775), and FIP200 (Proteintech; 1:2,000, 17250-1-AP). The following antibodies were used for ELISA-based recycling assays at a concentration of 5 µg/ml in 0.05 M $Na_2CO_3$, pH 9.6, at 4°C: MT1-MMP (MilliporeSigma; mab3328), $\alpha_5$-integrin (BD Pharmingen; 555651), and active β1 integrin (BD Biosciences; 09351D).

Plasmid expression vectors used included the following: mCherry-MT1-MMP, kindly gifted by Philippe Chavrier (Institut Curie, Paris, France), and CD63-GFP as previously described (Diaz-Vera et al., 2017). Expression vectors encoding PINK1-YFP (WT and kinase-dead; Lazarou et al., 2015) were kindly donated by Richard Youle (National Institutes of Health, Bethesda, MD). Rescue vectors for PINK1-YFP were generated by

introducing silent mutations in the region where PINK1 siRNA oligo 3 (Dharmacon; J004030-08-0005) binds. Mutations were introduced using the QuikChange Multi Site-Directed Mutagenesis Kit (Agilent; 210515) using the following primers: forward: 5′-GTACAGGGATAGTTCTTCATGACTAGAAATAGTGTCCGGCCATGGCCCAGGC-3′; and reverse: 5′-GCCTGGGCCATGGCCGGACACTATTTCTAGTCATGAAGAACTATCCCTGTAC-3′.

TLR9 reagents (InvivoGen) were all used at 1 µM: TLR9 agonist ODN 2006 (ODN 7909) and ODN 2006 control (ODN 2137), as well as TLR9 antagonist ODN TTAGGG (A151) and ODN TTAGGG control. Drug treatments were as follows with the working concentration in parentheses: group II glutamate metabotropic receptor agonist LY95 (100 nM; Tocris) and CCCP (10 µM; Sigma-Aldrich).

### siRNA, CRISPR, and transfections
siRNA oligonucleotides for Rab27, CD63, PINK1, FIP200, TLR9, ATG5, and ATG7 were all derived from SMARTpool (Dharmacon). For certain experiments (see Fig. 8 d), "stealth" siRNAs (Thermo Fisher Scientific/Life Technologies) were used to knock down TLR9. CRISPR/Cas9 knockout of PINK1 was performed as previously described (Novo et al., 2018), and the guide sequences were as follows: PINK1 1 CRISPR guide forward: 5′-CACCGCAAGCGCGTGTCTGACCCAC-3′; reverse: 5′-AAACGTGGGTCAGACACGCGCTTGC-3′; PINK1 2 CRISPR guide forward: 5′-CACCGTGTCGGACTTGAGATCCCGA-3′; reverse: 5′-AAACTCGGGATCTCAAGTCCGACAC-3′. Transient transfections of were performed using the AMAXA transfection system (Lonza) according to the manufacturer's instructions. Approximately 5 × $10^6$ cells were used for each transfection with 100 µl AMAXA solution V plus 5 µl siRNA reagent at 20 µM, or with 0.5 µg plasmid construct, and they were electroplated using the X-013 program. For overexpression of PINK1, cells were transfected with 2 µg expression vector. For rescue experiments, 2 µg of the appropriate expression vector was combined with 5 µl of 20 µM PINK1 3 siRNA (Dharmacon).

### qPCR and long-range PCR
qPCR and long-range PCR analysis was performed on DNA extracted from cells/EVs/liposomes using the GeneJET Genomic DNA Purification Kit (Thermo Fisher Scientific) with primers obtained from Invitrogen. Human gene primers used were as follows: ND1 forward: 5′-CGAAAGGACAAGAGAAATAAGC-3′; ND1 reverse: 5′-CTGTAAAGTTTTAAGTTTTATGCG-3′; β-globin forward: 5′-CCACTTCATCCACGTTCACC-3′; β-globin reverse: 5′-GAAGAGCCAAGGACAGGTAC-3′; tRNA-leu forward: 5′-CACCCAAGAACAGGGTTTGT-3′; tRNA-leu reverse: 5′-TGGCCATGGGTATGTTGTTA-3′; COX1 forward: 5′-CCAATACCAAACGCCCCTCT-3′; COX1 reverse: 5′-AGAATGGGGTCTCCTCCTCC-3′; ND6 forward: 5′-AATCATACAAAGCCCCCGCA-3′; ND6 reverse: 5′-TGGGGTTAGCGATGGAGGTA-3′; CYB forward: 5′-ATCACTCGAGACGTAAATTATGGCT-3′; CYB reverse: 5′-TGAACTAGGTCTGTCCCAATGTATG-3′; and mtDNA nt positions 8339–9334 forward: 5′-CCACCCAACAATGACTAATC-3′; mtDNA nt positions 8339–9334 reverse: 5′-GTATGAGGAGCGTTATGGAG-3′. Mouse gene primers used were as follows: ND1 forward: 5′-CCTATCACCCTTGCCATCAT-3′; ND1 reverse: 5′-GAGGCTGTTGCTTGT

GTGAC-3′; and genomic DNA forward: 5′-CAGCGCTAGAGTGAG CGGATTA-3′; genomic DNA reverse: 5′-GAATTGATCAGTCTG CAGACAGCC-3′.

qPCRs were prepared using the SYBR Green qRT-PCR kit (Qiagen), and amplified products were analyzed by a CFX96 Touch Real-Time PCR Detection System (Bio-Rad Laboratories) and CFX manager software from which the comparative cycle threshold was determined and presented as the fold change relative to control.

For long-range PCR, 100 ng of each sample was amplified in a 25-μl reaction containing 0.5 μl Takara PrimerSTAR GXL DNA polymerase and 5 μl of the corresponding buffer, 2 μl deoxynucleoside triphosphate, 0.5 μl mtDNA forward sequence 5′-TTAAAACTCAAAGGACCTGGC-3′ (10 μM), and 0.5 μl reverse primer sequence 5′-AGGGTGATAGAGCTGTGATC-3′ (primer stock concentration 10 μM). DNA was amplified under the following conditions: 94°C for 1 min, then 30 cycles of 98°C for 10 s and 68°C for 15 min, followed by 72°C for 10 min. The product was run at 45 V for 3 h in 0.8% agarose gel alongside lambda HindIII digest ladder as per the manufacturer's instructions (New England Biolabs).

### EV purification, NTA, and sucrose density gradient centrifugation

MDA-MB-231 cells or primary mouse mammary tumor cells (after transfection with siRNA if required) were plated into 15-cm tissue culture dishes at $2 \times 10^6$ cells/dish. Where indicated, cells were fed with glutamine-free media, and $4 \times 10^6$ cells/dish were plated. After 48 h, cells were washed with PBS, and 15 ml of EV-free DMEM was added. EV-free media was obtained by overnight centrifugation at 100,000 $g$. After 48 h, EVs were collected via differential centrifugation from the media as described previously (Novo et al., 2018). The volume of 0.02-μm filtered PBS used to resuspend final EV pellets depended on the original volume of conditioned media as follows: For instance, 22 μl of filtered PBS was used per 15-cm dish from which the EVs were collected. All EV isolations were stored at 4°C and used within 48 h.

NTA was performed using the NanoSight LM10 instrument (Malvern Panalytical) according to the manufacturer's instructions. Resuspended EVs were diluted 1:50 in filtered PBS before being introduced into the instrument for measurement. Sucrose density gradient centrifugation was performed as described previously (Novo et al., 2018).

### OCRs and mitochondrial mass measurements

OCRs were determined using the Seahorse Extracellular Flux Analyzer (Agilent) according to the manufacturer's instructions. Briefly MDA-MB-231 cells were plated at $3 \times 10^4$ cells/well in a 96-well plate, and assay medium was prepared with phenol-free DMEM. OCR was measured during stepwise injection of 1 μM oligomycin, 1 μM carbonyl cyanide-4-(trifluoromethoxy)phenylhydrazone (FCCP), and 1 μM rotenone/antimycin A (Sigma-Aldrich). After completion, cells were lysed with radioimmunoprecipitation assay buffer, and OCR was normalized to protein content determined using a Pierce bicinchoninic acid protein assay kit (Thermo Fisher Scientific) according to the manufacturer's instructions. Basal

and maximal OCRs were calculated by subtracting the average post-rotenone/antimycin A OCR value from the average initial OCR value (for basal respiration) and the post-FCCP value (for maximal respiration). To calculate basal OCR, the mean post-rotenone/antimycin A values were subtracted from the mean preoligomycin values. For maximum respiration rate, the mean post-rotenone/antimycin A values were subtracted from the mean post-FCCP values.

To calculate mitochondrial mass, cells were seeded at $3 \times 10^5$ cells/well in six-well tissue culture dishes, and, after 16 h, they were pretreated with the fluorescent mitochondrial probe MitoTracker Green (100 nM; Thermo Fisher Scientific) for 30 min, and the mean fluorescence intensity per cell was determined using FACS. Cells were detached by trypsin and washed in PBS before resuspension in 1 ml PBS and transferred to FACS tubes (STEMCELL Technologies). Samples were imaged using the FACSCalibur machine (BD Biosciences) and analyzed using FlowJo single-cell analysis software to determine the average per-cell fluorescence intensity.

### Measurement of mitochondrial membrane potential

MDA-MB-231 cells were treated with LY95 (for 3 or 5 d) and plated into six-well dishes ($3 \times 10^5$ cells/well). Cells were then incubated overnight in the presence or absence of CCCP (10 μM). Media were then supplemented with tetramethylrhodamine ethyl ester (TMRE; 50 μM, ab113852; Abcam) for 30 min at 37°C. Cells were then detached by trypsinization, resuspended in 1 ml PBS containing DAPI and analyzed using flow cytometry. Data were analyzed using FlowJo software with gates positioned to remove duplicate cells and dead cells (via positive DAPI signal). The mean fluorescence of TMRE was then plotted as the fold change compared with untreated control.

### TIRF microscopy

MDA-MB-231 cells pretreated with EVs or liposomes were transfected with mCherry-MT1-MMP plasmid plated directly onto 3.5-cm glass-bottomed plates, which were precoated with fibronectin (Sigma-Aldrich), diluted in PBS to 50 μg/ml. After 18 h, the cell media were changed to DMEM lacking glutamine plus treatments including EVs, liposomes, and/or drug treatments for at least 1 h before imaging in a 5% $CO_2$ atmosphere at 37°C using either a 100× or 60× plan apochromat TIRF objective on a Nikon Eclipse Ti microscope. Images were collected every 1 s for 5 min as time-lapse videos and analyzed in ImageJ using the TrackMate plugin (Tinevez et al., 2017) to determine the appearance of fluorescent puncta per minute of each video. Kymographs were obtained via the orthogonal view of each video adjusted to 40 μm.

### Confocal fluorescence microscopy

2 d before imaging, MDA-MB-231 cells were transfected with DSRed-mito (Clontech) and GFP-CD63 expression plasmids with either nontargeting control or PINK1 targeting SMARTpool siRNAs (Dharmacon). The following day, cells were detached and plated onto fibronectin-coated 3.5-cm glass-bottomed dishes. At least 3 h before imaging, media were changed to full DMEM. CCCP (10 μM) or DMSO was added 1 h before image collection.

Images were acquired on a Zeiss LSM 880 Airyscan confocal microscope in Airyscan Fast mode. To minimize bleed-through,

channels were imaged separately. Cells were imaged in an environmental chamber maintained at 37°C and 5% $CO_2$. For each time-lapse series, 500 frames were collected with an interval of 0.59 s. Images were then processed using the Airyscan processing function of the ZEN software.

For each distinct mitochondrion, the mitochondrial area encompassed at the beginning of the video (t = 0 s), together with the number and duration of contacts made with CD63-GFP–positive endosomes, was recorded. For each cell, all mitochondria whose envelope could be distinguished at t = 0 were analyzed, and values were normalized to the area of each mitochondrion. ImageJ Fiji software was used for measurements. Perinuclear mitochondria and CD63-GFP–containing endosomes formed large static clumps that could not easily be distinguished. These were unaltered by transfection with PINK1-targeting siRNA oligonucleotides and were omitted from analysis.

## Invasion assays using cell-derived matrices and fluorescent gelatin

For pseudopod invasion assays, MDA-MB-231 cells were seeded at $8 \times 10^4$ cells/well into fibroblast-derived matrices made as described previously (Cukierman et al., 2001) using human immortalized fibroblasts prepared as described elsewhere (Timpson et al., 2011). 4 h later, cells were imaged using time-lapse phase-contrast microscopy (Axiovert S100; Carl Zeiss Microscopy; 10× objective) in an atmosphere of 5% $CO_2$ at 37°C as described previously (Rainero et al., 2012). Pseudopod lengths were measured using ImageJ software. Fluorescent gelatin degradation assays were performed as described previously (Dornier et al., 2017).

## ELISA-based recycling assays

MDA-MB-231 cells were treated with ~900,000 liposomes/10 ml media (without glutamine) for at least 24 h. Then cells were incubated in serum-free DMEM in the absence of glutamine for 1.5 h, transferred to ice, washed twice in cold PBS, and surface labeled at 4°C with 0.2 mg/ml NHS-SS-biotin [succinimidyl 2-(biotinamido)-ethyl-1,3′-dithiopropionate; Pierce] in PBS for 30 min. Cells were transferred to serum-free and glutamine-free DMEM for 30 min at 37°C to allow internalization of the tracer. Cells were returned to ice, washed twice with ice-cold PBS, and biotin was removed from proteins remaining at the cell surface by reduction with MesNa (2-mercaptoethanesulfonate). The internalized fraction was then chased from the cells by returning them to 37°C in serum-free DMEM. At the indicated times, cells were returned to ice, and biotin was removed from recycled proteins by a second reduction with MesNa. Biotinylated MT1-MMP, active β1, and α5 integrin were then determined by capture ELISA using MaxiSorp (Nunc) plates coated with antibodies recognizing each receptor as described previously (Dozynkiewicz et al., 2012).

## mtDNA-deficient MDA-MB-231 cells

MDA-MB-231 cells with reduced mtDNA content (Rho0) were generated as previously described (Giampazolias et al., 2017). Briefly, cells were treated with ethidium bromide (500 ng/ml) plus uridine (50 µg/ml) for 5 d. Following this, cells were incubated in EV-free media for EV collection over a 48-h period.

## DNase beads and liposome preparation

DNase beads were made using Affi-Gel 10 beads (Bio-Rad Laboratories) and DNase I lyophilized powder (Qiagen). 10 ml of Affi-Gel 10 bead slurry was washed via a vacuum pump with >400 ml ice-cold water followed by >300 ml coupling buffer (0.1 M Hepes, 80 mM $CaCl_2$, pH 7.2), then the volume was reduced to 8 ml. DNase I, reconstituted in 4 ml coupling buffer, was added to 8 ml resin and incubated on a roller at 4°C overnight. Then the beads were washed with >400 ml coupling buffer with 1 M NaCl followed by >400 ml coupling buffer. Following this, the slurry was reduced to ~4 ml, then supplemented with 4 ml coupling buffer and 4 ml glycerol and kept at –20°C in 500-µl aliquots.

45-µl fractions of 1-palmitoyl-2-oleoyl-glycero-3-phosphocholine (POPC) lipids (25 mg/ml; Avanti Polar Lipids) were dried in glass tubes for 1–2 h in a Genevac centrifugal evaporator to remove all traces of chloroform. Lipids were then stored at –20°C or used directly. Dried lipids were resuspended in 500 µl PBS to achieve a concentration of 3 mM. For encapsulated mtDNA, lipids were resuspended in PBS containing 1,000 ng mtDNA (previously extracted from MDA-MB-231 cells using an mtDNA isolation kit; Abcam). Resuspended lipids were then sonicated in a water bath for 10 min at 45°C and then subjected to five rounds of snap freezing/thawing using liquid nitrogen. Lipids were then extruded through a 100-nm filter at RT using a Mini Extruder (Avanti Polar Lipids), and this was repeated 21 times. Extruded lipids were stored for up to 2 wk at 4°C, and their size and number were assessed using NTA.

To remove mtDNA not encapsulated by liposomes, 25 µl prepared liposomes were added to 50 µl DNase resin, 20 µl $MgCl_2$ buffer (50 mM Tris, pH 7.5, 12.5 mM $MgCl_2$, 2.5 mM CaCl) and PBS, then incubated at 37°C at 450 rpm in a thermomixer (Eppendorf) for 1 h. Following this, the DNase-conjugated beads were precipitated by centrifugation, and 100 µl supernatant (containing the liposome-encapsulated mtDNA) was stored for up to 2 wk at 4°C.

## Immunoelectron microscopy

For immunolabeling, EVs isolated by differential centrifugation were fixed using PFA (4%). An aliquot (10 µl) of fixed EV suspension was placed onto a 100-mesh formvar-carbon–coated grid for 20 min to allow the EVs to adhere. Grids were washed in PBS, and the nonspecific binding site was blocked using BSA (3% in PBS) for 1 h. Adherent EVs were then incubated sequentially with an antibody recognizing CD63 and a secondary antibody conjugated to 10-nm gold particles (Aurion) for 1 h. For contrast, grids were then incubated in a mixture of oxalate/uranyl acetate for 10 min in the dark, then in methylcellulose/uranyl acetate for 10 min on ice in the dark, and allowed to dry in metal loops overnight. The grids were imaged using a JEOL 1200 EX transmission electron microscope operating at 80 kV. All acquired images were processed (brightness and contrast), and all EV diameters were determined, using Fiji software (Schindelin et al., 2012).

## Online supplemental material

Fig. S1 describes the consequences of siRNA of Rab27 on release of EVs from MDA-MB-231 cells, and these data pertain to Fig. 1. Fig. S2 displays data outlining the consequences of siRNA of Rab27 and CD63 on oxygen consumption by MDA-MB-231 cells, and this pertains to Fig. 2 a. Fig. S3 displays an analysis of mitochondrial genes within EVs that pertains to Fig. 2, b–f. Fig. S4

displays Western blot and oxygen consumption data that pertain to Fig. 3. Fig. S4 also displays data illustrating the consequences of CRISPR knockout of PINK1 in cells from the MMTV-PyMT mouse model of mammary carcinoma. Fig. S5 contains data indicating that disruption of autophagy regulators (ATG5/7 or FIP200) does not impact EV release from MDA-MB-231 cells. Video 1 shows a clip of the dynamics of late endosomes and mitochondria in control cells. Video 2 shows a clip of the dynamics of late endosomes and mitochondria in PINK1 knockdown cells. Video 3 shows a clip of the dynamics of late endosomes and mitochondria in CCCP-treated cells. Video 4 shows a clip of the dynamics of late endosomes and mitochondria in CCCP-treated PINK1 knockdown cells. Table S1 shows the catalog numbers of the antibodies used in the study.

### Data availability

The data supporting the findings of this study are available within the article and its supplementary information files and from the corresponding author upon request.

## Acknowledgments

We thank Richard Youle (National Institutes of Health, Bethesda, MD) and Chunxin Wang (Institut Curie, Paris, France) for the PINK1-YFP constructs, Anh Hoang Le (Machesky laboratory, Cancer Research UK Beatson Institute) for help with the gelatin degradation protocol, and Jaclyn Long (Ryan laboratory, Cancer Research UK Beatson Institute) for advice with LC3 blots.

This work was funded by Cancer Research UK and Breast Cancer Now. We acknowledge the Cancer Research UK Glasgow Centre (C596/A18076) and the Biostatistics Unit facilities at the Cancer Research UK Beatson Institute (C596/A17196).

The authors declare no competing financial interests.

Author contributions: N. Rabas, S. Palmer, L. Mitchell, L. Lemgruber Soares, and J.C. Norman performed the experiments and analyzed experimental data. J.C. Norman supervised the experimental work. S. Ismail, A. Gohlke, J.S. Riley, S. Tait, P. Gammage, and I.R. Macpherson contributed to the conceptual design and planning of experiments and provision of reagents. N. Rabas, S. Palmer, and J.C. Norman prepared the figures and wrote the manuscript.

Submitted: 9 June 2020

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

# Supplemental material

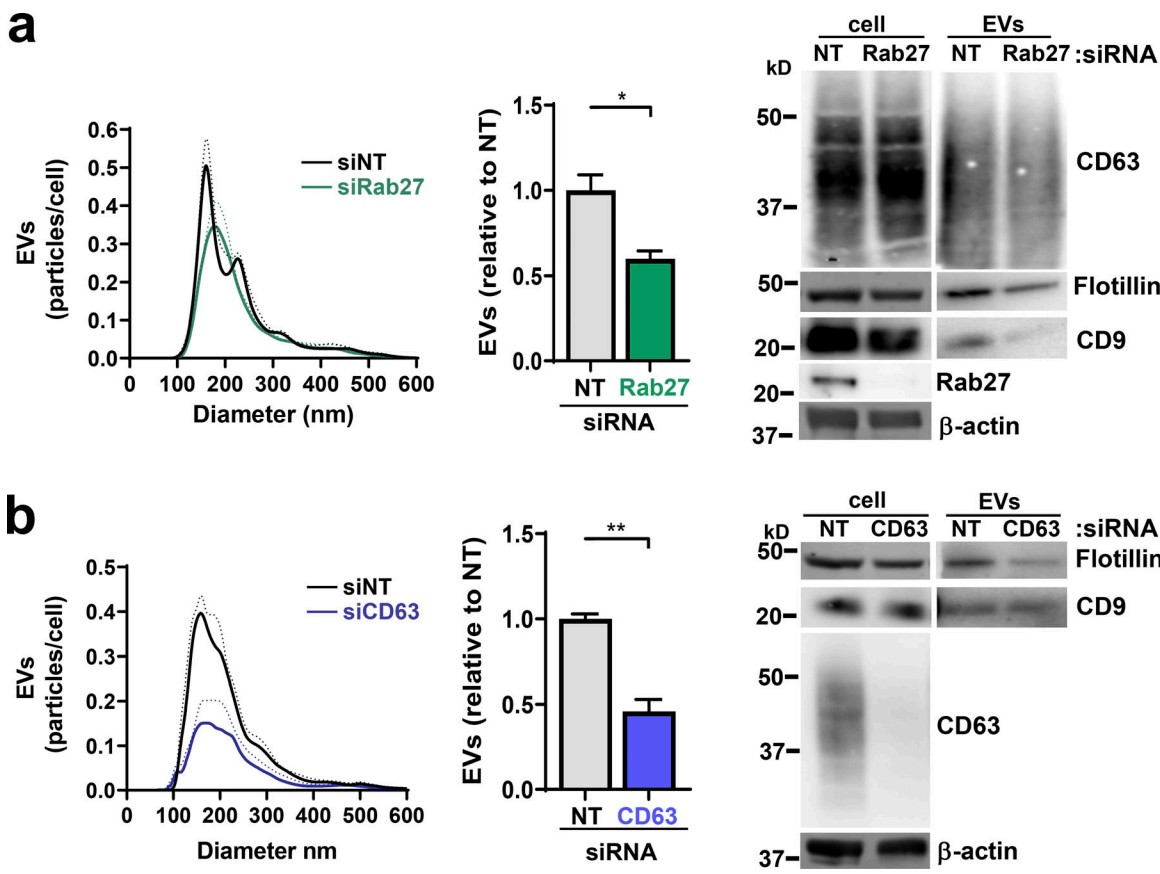

Figure S1. **EVs are released in a Rab27- and CD63-dependent manner. (a and b)** MDA-MB-231 cells were transfected with siRNAs targeting Rab27a/b, CD63, or nontargeting (NT) siRNA. Cells were incubated for 48 h in glutamine-replete medium, and EVs were isolated by differential centrifugation before analysis by nanoparticle tracking to obtain size and concentration profiles. Values were normalized to cell number and are expressed as the number of particles of a given size, in 0.5-nm increments, per cell. Values represent the mean of $n$ = 4 independent experiments; dotted line and error bars represent SEM. **, P < 0.01; *, P < 0.05; $t$ test with Welch's correction. Cell lysates and differential centrifugation pellets were also analyzed by Western blotting for the indicated EV markers. Actin was used as a loading control.

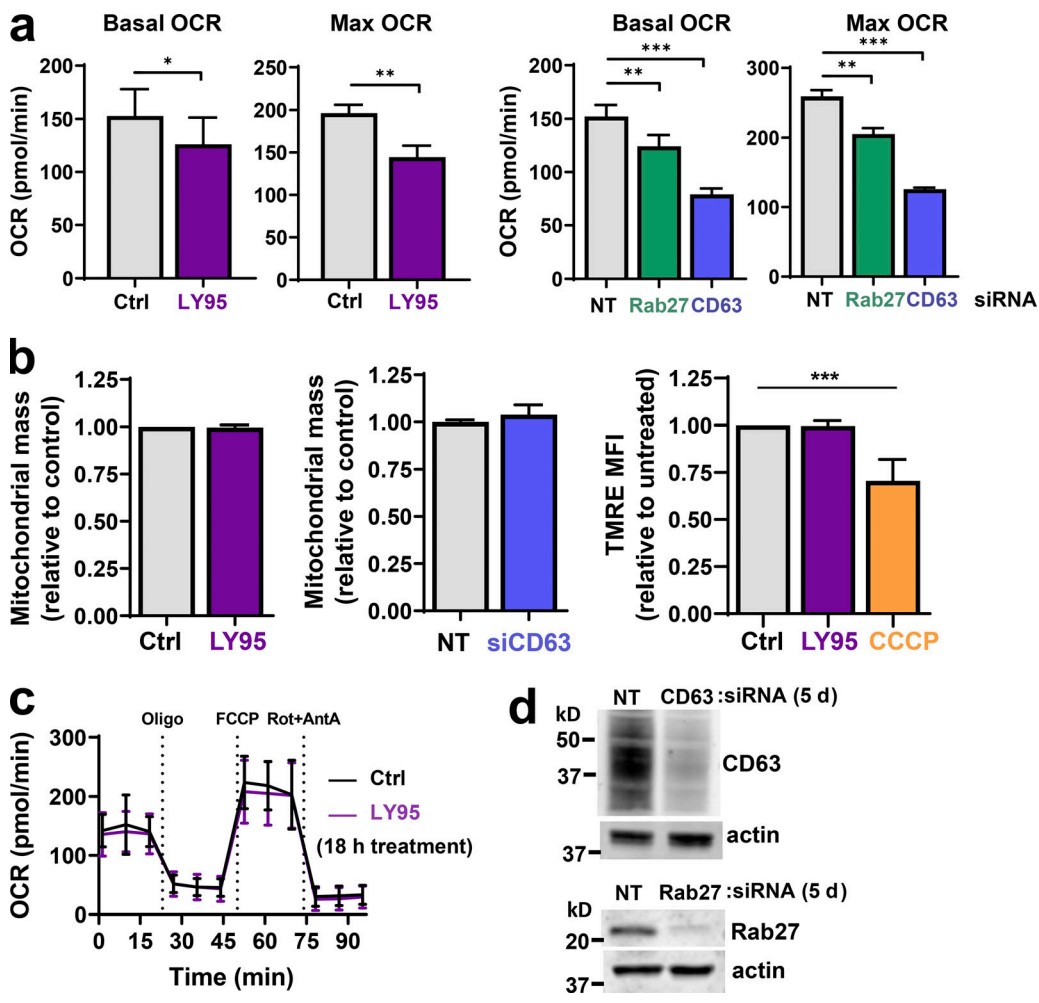

Figure S2. **mGluR3, CD63, and cellular oxygen consumption. (a–c)** MDA-MB-231 cells were transfected with siRNAs targeting Rab27a/b, CD63, or non-targeting (NT) siRNAs or were left untransfected and incubated in the presence (LY95) or absence (Ctrl) of LY95 (0.3 µM) for 5 d (a and b) or 18 h (c). Oxygen consumption was determined using the Seahorse XF$^e$96 Extracellular Flux Analyzer. Values represent mean ± SEM of three independent experiments. The basal and maximal (Max) OCRs were extracted from data such as those displayed in Fig. 2 a. Maximum OCR was determined by subtracting rotenone/antimycin A (Rot+AntA) OCR from basal OCR values. Values represent the mean of three independent experiments; error bars signify SEM. *, P < 0.033; **, P < 0.01; ***, P < 0.001; paired Student's *t* test. Mitochondrial mass was determined using the fluorescent mitochondrial probe MitoTracker Green, and the mean fluorescence intensity per cell was quantified using flow cytometry; values are mean ± SEM of *n* = 3 (b). TMRE (50 nM) was added to cells for 30 min before analysis by flow cytometry, and CCCP was added for 16 h before this as a positive control for loss of TMRE fluorescence through uncoupling of the mitochondria. Median fluorescence intensities (MFIs) are plotted relative to control; values are SEM of *n* = 3. **(d)** Western blotting was used to confirm the effectiveness of siRNAs for CD63 and Rab27 5 d following transfection with actin used as a loading control.

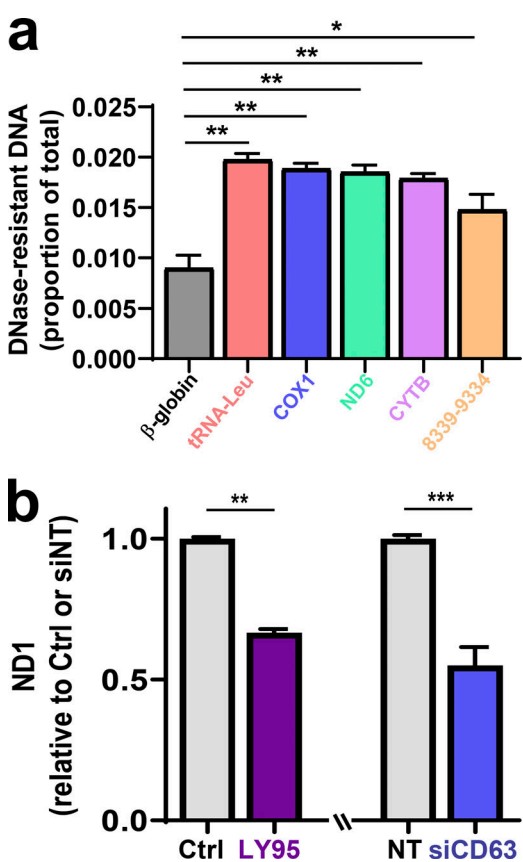

Figure S3.   **mtDNA is protected within EVs. (a)** EVs were purified from MDA-MB-231 cell-exposed medium by differential centrifugation and then incubated in the presence and absence of DNase, which had been immobilized on agarose beads. Following removal of the DNase-conjugated beads by centrifugation, the quantity of mtDNA was determined relative to EVs not treated with DNase using qPCR with primers complementary to sequences within the indicated mitochondrial genes and the β-globin nuclear gene. Bars represent the fold change relative to control. Values are the mean ± SEM. **(b)** MDA-MB-231 cells were transfected with siRNAs targeting CD63 (siCD63) or a nontargeting siRNA (siNT) or were left untransfected. Cells were then incubated for 48 h in the absence (Ctrl; NT and siCD63) or presence of 0.3 µM LY95, and medium was collected over this period. EVs were purified from the medium by differential centrifugation as above, and half of the sample was treated with DNase-conjugated beads. The mtDNA content of these samples was determined using qPCR with primers complementary to a sequence within the mitochondrial ND1 gene. Values represent the mean fold change relative to control from three independent experiments. Error bars represent SEM. **, P < 0.002; Mann-Whitney *U* test.

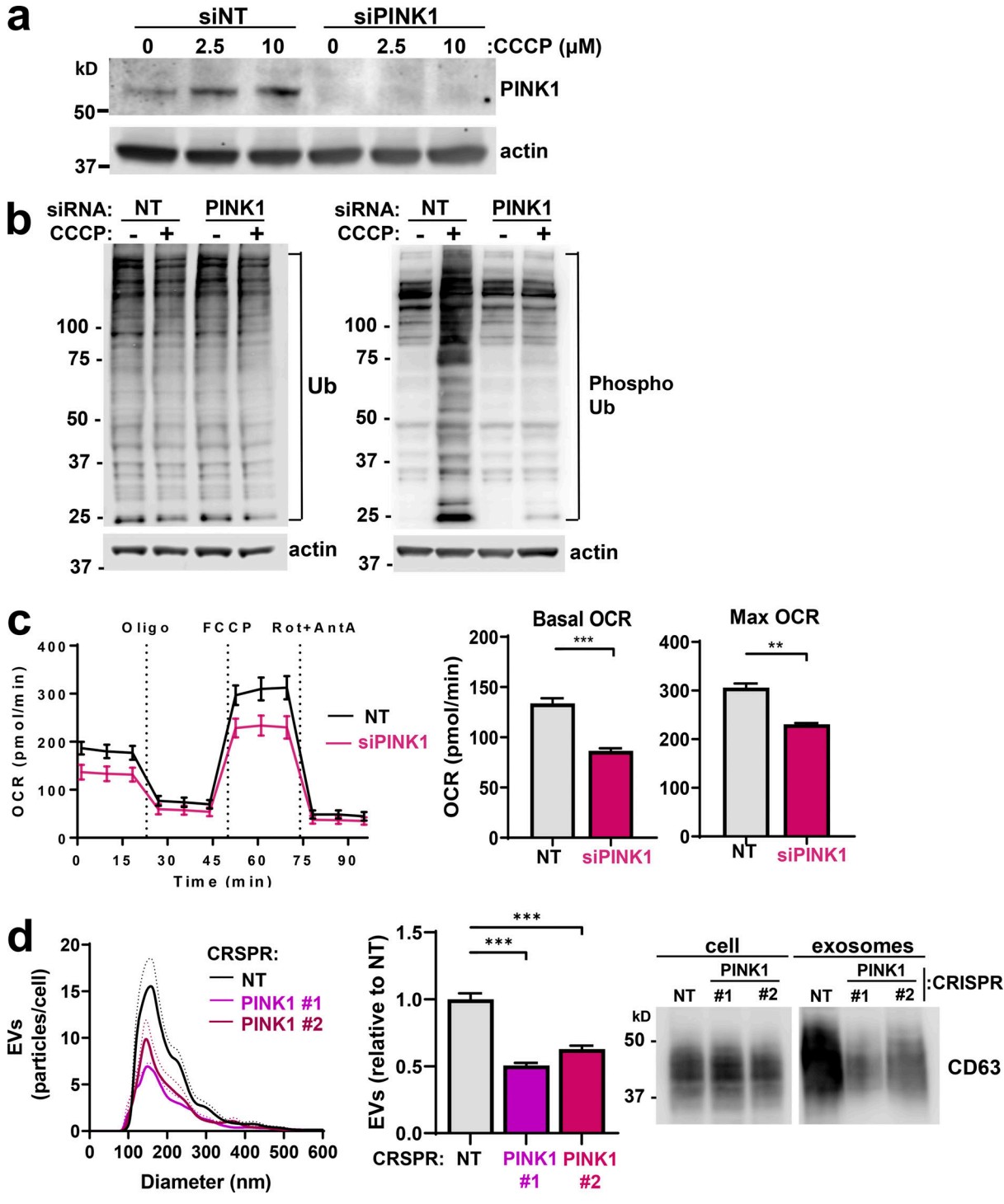

Figure S4. **PINK1 in mitochondrial function and EV production. (a)** MDA-MB-231 cells were transfected with siRNAs targeting PINK1 (siPINK1) or a nontargeting control (siNT). Transfected cells were treated with the indicated concentrations of CCCP for 3 h, and PINK1 levels were determined using Western blotting with actin as a loading control. **(b)** MDA-MB-231 cells were transfected siRNAs targeting PINK1 (siPINK1) or nontargeting control (siNT) and treated with CCCP for 3 h before being analyzed for total Ub and phosphorylated Ub by Western blotting. Actin was used as a loading control. **(c)** MDA-MB-231 cells were transfected with siRNAs targeting PINK1 (siPINK1) or nontargeting (NT) siRNA. 5 d following transfection, oxygen consumption was determined using the Seahorse XFe96 Extracellular Flux Analyzer. Sequential treatment with oligomycin, FCCP, and rotenone/antimycin A (Rot+AntA) was performed as indicated. Maximum OCR was determined by subtracting rotenone/antimycin A OCR from basal OCR values. Values represent the mean of three independent experiments; error bars signify the SEM. **, P < 0.01; ***, P < 0.001; paired Student's t test. **(d)** Conditioned medium was collected from cells isolated from an MMTV-PyMT–driven mammary carcinoma in which PINK1 expression had been disrupted by CRISPR using two distinct guide sequences (PINK1 no. 1 and PINK1 no. 2) or a nontargeting guide sequence (NT), and EVs were purified from these by differential centrifugation. EVs were analyzed using nanoparticle tracking and Western blotting for CD63 as in Fig. 1 a. Values represent the mean of three independent experiments; dotted line and error bars represent SEM. ***, P < 0.001; t test with Welch's correction.

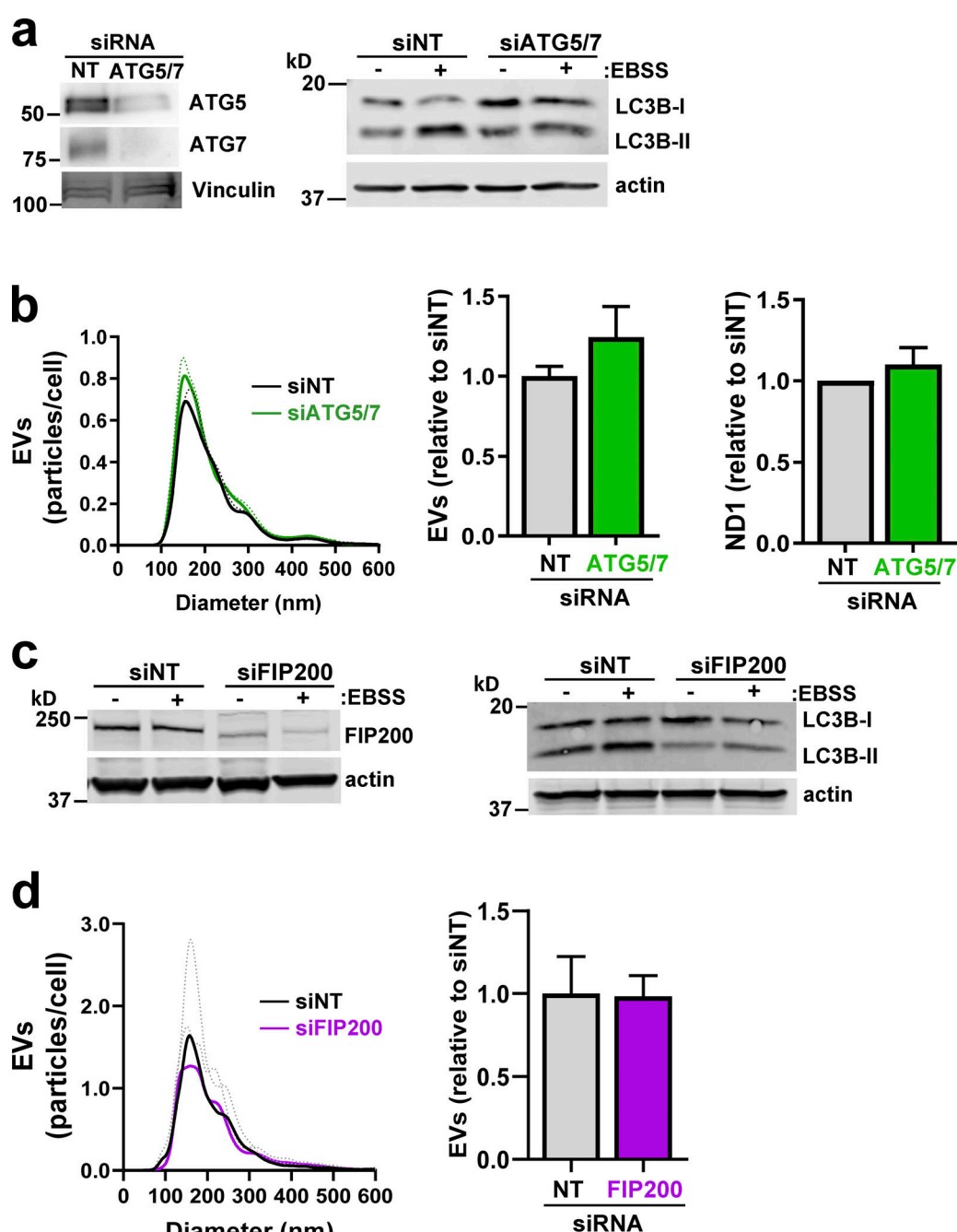

Figure S5. **Inhibition of mitophagy does not oppose release of mtDNA-containing EVs. (a–d)** MDA-MB-231 cells were transfected with siRNAs targeting ATG5 and ATG7 (siATG5/7; a and b), FIP200 (siFIP200; c and d), or a nontargeting control (siNT). Conditioned medium was collected from transfected cells over a 48-h period. EVs were purified from conditioned medium using differential centrifugation and analyzed by nanoparticle tracking. mtDNA was determined using qPCR with primers complementary to a sequence within the mitochondrial ND2 gene. Values represent the mean of three independent experiments; dotted line and error bars represent SEM. Cellular levels of ATG5/7 (a), FIP200 (c), and LC3B (a and c) were determined using Western blotting with actin or vinculin as a loading control. In a and c, cells were nutrient starved in Earle's balanced salt solution (EBSS) where indicated before lysis.

Video 1. **Dynamics of late endosomes and mitochondria in control cells.** MDA-MB-231 cells were transfected with GFP-tagged CD63 (GFP-CD63; green) to visualize late endosomes and mitochondria-targeted DsRed (DsRed-mito; red) in combination with a nontargeting siRNA (siNT). Transfected cells were plated onto glass-bottomed dishes, incubated in the presence of vehicle control (DMSO), and imaged using a Zeiss LSM 880 Airyscan confocal microscope at 37°C with frames collected every 0.59 s. The frame display rate is 30 frames/s⁻¹. Representative stills from, and quantitation of, this video are presented in Fig. 5.

Video 2.   **Dynamics of late endosomes and mitochondria in PINK1 knockdown cells.** MDA-MB-231 cells were transfected with GFP-tagged CD63 (GFP-CD63; green) to visualize late endosomes and mitochondria-targeted DsRed (DsRed-mito; red) in combination with an siRNA targeting PINK1 (siPINK1). Transfected cells were plated onto glass-bottomed dishes, incubated in the presence of vehicle control (DMSO), and imaged using a Zeiss LSM 880 Airyscan confocal microscope at 37°C with frames collected every 0.59 s. The frame display rate is 30 frames/s$^{-1}$. Representative stills from, and quantitation of, this video are presented in Fig. 5.

Video 3.   **Dynamics of late endosomes and mitochondria in CCCP-treated cells.** MDA-MB-231 cells were transfected with GFP-tagged CD63 (GFP-CD63; green) to visualize late endosomes and mitochondria-targeted DsRed (DsRed-mito; red) in combination with a nontargeting siRNA (siNT). Transfected cells were plated onto glass-bottomed dishes, incubated in the presence of the mitochondrial uncoupler CCCP (10 μM), and imaged using a Zeiss LSM 880 Airyscan confocal microscope at 37°C with frames collected every 0.59 s. The frame display rate is 30 frames/s$^{-1}$. Representative stills from, and quantitation of, this video are presented in Fig. 5.

Video 4.   **Dynamics of late endosomes and mitochondria in CCCP-treated PINK1 knockdown cells.** MDA-MB-231 cells were transfected with GFP-tagged CD63 (GFP-CD63; green) to visualize late endosomes and mitochondria-targeted DsRed (DsRed-mito; red) in combination with an siRNA targeting PINK1 (siPINK1). Transfected cells were plated onto glass-bottomed dishes, incubated in the presence of the mitochondrial uncoupler CCCP (10 μM), and imaged using a Zeiss LSM 880 Airyscan confocal microscope at 37°C with frames collected every 0.59 s. The frame display rate is 30 frames/s$^{-1}$. Representative stills from, and quantitation of, this video are presented in Fig. 5.

**Provided online is one table. Table S1 shows the catalog numbers of the antibodies used in the study.**

