## [Peer Review File · The Journal of Cell Biology]

PINK1 drives production of mtDNA-containing extracellular vesicles to promote invasiveness

Nicolas Rabas, Sarah Palmer, Louise Mitchell, Shehab Ismail, Andrea Gohlke, Joel Riley, Stephen Tait, Payam Gammage, Leandro Lemgruber, Iain Macpherson, and Jim Norman

Corresponding Author(s): Jim Norman, CRUK-Beatson Institute

Review Timeline:

Submission Date:	2020-06-09
Editorial Decision:	2020-07-07
Revision Received:	2021-07-29
Editorial Decision:	2021-08-26
Revision Received:	2021-09-09

Monitoring Editor: Kenneth Yamada

Scientific Editor: Andrea Marat

Transaction Report:

DOI: <https://doi.org/10.1083/jcb.202006049>

July 7, 2020

Re: JCB manuscript #202006049

Prof. Jim C Norman
CRUK-Beatson Institute
Integrin Cell Biology Lab
Beatson Institute for Cancer Research Garscube Estate
Switchback Road
Glasgow, East Dumbartonshire G61 1BD

Dear Jim,

Thank you for submitting your manuscript entitled "Glutaminolysis promotes PINK1-dependent production of mitochondrial DNA-containing exosomes to drive invasion". The manuscript has now been assessed by three expert reviewers; their reports are appended below. As you can see from the reviews provided by three leaders in the various overlapping research areas spanned by this paper, there was interest in the conclusions, but also some major concerns about the strength of the documentation that must be resolved. After a careful assessment of the feedback from the expert reviewers, who differed in their levels of enthusiasm for this study, we feel that the significant concerns raised by the reviewers unfortunately preclude publication of the current version of the manuscript in JCB.

These constructive peer reviewers raise various concerns about the strength of the experimentation underlying the various intriguing conclusions and have provided a series of specific suggestions. We feel that it would be particularly important to provide additional, more direct experimental support for the key current conclusions. Significant points appear to us to include measurement of mitochondrial membrane potential and PINK1 rescue in the siRNA experiments with PINK1 overexpression to test effects on exosome formation. There were various requests for more (semi)quantification of western blots that appear reasonable. Two reviewers felt that the CCCP treatment data were not convincing, requiring some additional approach, and the evidence for mtDNA packaging into exosomes accounting for tumor invasiveness needs stronger evidence. There were also requests to document further the purity of the exosomes. Another general theme involved requests to provide more convincing data for manipulations that result in only partial effects. The reviewers have provided a variety of other suggestions and requests that we ask you to consider carefully and resolve if practical. Although exploring further the mechanisms of PINK1 action in late endosome/MVB contact would indeed be quite intriguing, it is not in our minds as essential as providing data supporting the current specific conclusions.

We realize that the revisions listed above will require substantial extra experimental work to satisfactorily address the concerns of the reviewers and meet the high standards of JCB, which we sincerely hope will be possible for you. We would be happy to further discuss the revisions if you have any questions or anticipate issues addressing these points. Please let us know if you are able to address the major issues outlined above and wish to submit a revised manuscript to JCB. A resubmitted manuscript would be again peer-reviewed to determine whether the main concerns were resolved.

As you may know, the typical timeframe for revisions is three to four months. However, we at JCB realize that the implementation of social distancing and shelter-in-place measures that limit the spread of COVID-19 also pose challenges to scientific researchers. Lab closures especially are preventing scientists from conducting experiments to further their research. Therefore, JCB has waived the revision time limit. We recommend that you reach out to the editors once your lab has reopened to decide on an appropriate time frame for resubmission. Because of the potential high interest of this study, we would be willing to re-review a manuscript that is delayed for other reasons if the findings remain sufficiently novel. Please note that papers are generally considered through only one revision cycle, so any revised manuscript will likely be either accepted or rejected.

If you choose to revise and resubmit your manuscript, please also attend to the following editorial points. Please direct any editorial questions to the journal office.

GENERAL GUIDELINES:

Text limits: Character count is < 40,000, not including spaces. Count includes title page, abstract, introduction, results, discussion, acknowledgments, and figure legends. Count does not include materials and methods, references, tables, or supplemental legends.

Figures: Your manuscript may have up to 10 main text figures. To avoid delays in production, figures must be prepared according to the policies outlined in our Instructions to Authors, under Data Presentation, <http://jcb.rupress.org/site/misc/ifora.xhtml>. All figures in accepted manuscripts will be screened prior to publication.

IMPORTANT: It is JCB policy that if requested, original data images must be made available. Failure to provide original images upon request will result in unavoidable delays in publication. Please ensure that you have access to all original microscopy and blot data images before submitting your revision.

Supplemental information: There are strict limits on the allowable amount of supplemental data. Your manuscript may have up to 5 supplemental figures. Up to 10 supplemental videos or flash animations are allowed. A summary of all supplemental material should appear at the end of the Materials and methods section.

If you choose to resubmit, please include a cover letter addressing the reviewers' comments point by point. Please also highlight all changes in the text of the manuscript.

Regardless of how you choose to proceed, we hope that the comments below will prove constructive as your work progresses. We would be happy to discuss them further once you've had a chance to consider the points raised. You can contact the journal office with any questions, cellbio@rockefeller.edu or call (212) 327-8588.

Thank you for thinking of JCB as an appropriate place to publish your work.

With kind regards,

Ken

Kenneth Yamada, MD, PhD
Editor, Journal of Cell Biology

Reviewer #1 (Comments to the Authors (Required)):

One main novel finding in this manuscript is that PINK1 mediates mitochondrial recruitment into exosomes by connecting mitochondria to late endosomes/multivesicular bodies. This occurs independent of the mitophagy role of PINK1 and requires the established exosome pathway component Rab27. A second major conclusion is that mtDNA in the exosomes, via TLR9, induces invasiveness of tumor cells. The authors cover a lot of ground in this manuscript. Unfortunately, many aspects are not rigorously explored and the results are open to many interpretations beyond the conclusions made.

Major issues

- 1) Although the decrease in mitochondrial proteins and DNA in exosomes following LY95 treatment is quite interesting, it is not expected that the absence of Cycophilin D protein would be greater than the absence of mtDNA (comparing Fig. 2b and 2e) in the exosomes. Also, it was already known that mitochondria are present in exosomes so the fewer exosomes in cells treated with the mGluR3 antagonist LY95 is around the same amount as the lower mitochondrial content (comparing Fig. 1a and Fig. 2f) is not unanticipated by the prior literature. The effect of LY95 on respiration may be very indirect (Fig. 2a).
- 2) Figure 3 presents a main novel finding, that PINK1 is required for exosome release. However, the rationale for addressing this is not clear. As the authors note, PINK1 senses mitochondrial damage and an almost complete loss of mitochondrial membrane potential is required for PINK1 stabilization. The authors do show a minor loss of respiration but need to measure mitochondrial membrane potential using TMRM or TMRE. Does exposure of cells to glutamate as in Fig. 1b affect mitochondrial membrane potential? Rescue of the PINK1 siRNA is required in Fig. 3a regarding siPINK inhibition of exosome production and also for Fig. 3d.
- 3) The authors state that phospho-ubiquitin is highly abundant in exosomes based on Suppl. Fig. 3c. What they show is phospho-ubiquitin is detectable. How do the authors quantify phospho-ubiquitin to determine it is highly abundant? Relative to what? Total ubiquitin?
- 4) Using CCCP to test the effect of increased levels of PINK1 on exosome formation is very crude and susceptible to many off target effects of CCCP (such as Opa1 loss and thus mitochondrial fragmentation into smaller vesicles). Overexpression of PINK1 should be performed instead.
- 5) The authors nicely show that CCCP stabilizes PINK1 in Suppl. Fig. 2a. Western blots of PINK1 are also needed for the siRNA experiments (in addition to the nice phosphoUb experiments in Suppl. 2b). Most importantly, western blots of PINK1 are needed during exosome formation and when exosome production is inhibited by LY95.
- 6) The authors indicate that PINK1 is required for mitochondrial contact with late endosomes (Fig. 3d). Does this contact require PINK1 enzyme activity? The authors should compare mitochondrial contact with late endosomes after rescuing siPINK1 treated cells with kinase dead PINK1 relative to rescue with wild type PINK1. More mechanistic insight into how PINK1 mediates late

endosome/multivesicular body contact is a key issue to address.

7) The authors then show that "donor" exosomes influence the trafficking of MT1-MMP, a cell surface receptor involved in tumor invasion, and that inhibition of the DNA sensor, TLR9, or PINK1 inhibits trafficking of this marker of tumor cell invasiveness. To assess if mitochondrial DNA is required for tumor invasion, the authors use a very indirect approach. Liposomes containing mtDNA are incubated with MDA-MB-231 cells and shown to cause increased MT1-MMP trafficking to the ventral cell surface. This activity required TLR9 activity. Although interesting, this does not indicate mtDNA packaging into exosomes is required or involved in tumor invasiveness. A loss of mtDNA approach is needed. Producing Rho0 exosome donor cells and assessing activity of their exosomes would be one rigorous approach to test the conclusion.

Minor point: At the end of page 6 the mention of fig 2a is incorrect.

Reviewer #2 (Comments to the Authors (Required)):

In the present manuscript, Rabas et al., describe a mechanism of intercellular communication that promote invasive behavior of breast cancer cells through a metabolic circuitry, requiring three mechanistic events: 1) mitochondrial damage in cancer cells activates PINK1 and leads to mitochondria-endosome proximity, possibly facilitating the inclusion of mtDNA in exosomes; 2) the upregulation of extracellular glutamate activates mGluR3 receptor that in turn leads to the Rab27-dependent exocytosis of mtDNA-containing endosomes; 3) the mtDNA activates TLR9-dependent pro-invasive behavior in the recipient cells. Though a combination of biochemical and molecular genetic experiments, as well as liposome-based reconstitution assays, the authors demonstrate that mtDNA in exosomes is sufficient to activate the TLR9-dependent invasive matrix-degradation program in the recipient cells.

The work deals with a relevant issue in cancer biology, as it describes a mechanism through which cancer cells can cope with metabolic stressful environments and get advantage from them. Experiments are very well conducted and carefully controlled. The molecular mechanism described is well dissected. For all the above reasons, I suggest this work for publication after some revisions, as indicated below.

- 1) Figure 2D. The protection from DNAase of mtDNA as compared to beta-globin DNA seems not very convincing. There's only 30% residual mtDNA after treatment vs. 15% of beta-globin. Can the authors further comment and discuss this point? Is this difference biologically and statistically significant?
- 2) Figure 2E. Please also provide Rab27 KD to confirm the proposed mechanism.
- 3) Figure 3D. The results shown here could be biased by the fact that CCCP treatment caused fragmentation of the mitochondria and this might affect the quantitation of contacts number. Can the authors further discuss this point in the text? In addition, can they also explain and further discuss why CCCP treatment does not increase the number of contacts, but only their duration?
- 4) Figure 4C. Please provide the effect of RAB 27 KD on gelatin degradation assay.
- 5) Figure 5D-E. To prove the in trans cell-to-cell communication, authors should perform the TLR9 KD in the recipient cell providing exosomes produced by control donor cells (as shown in the scheme in panel A).
- 6) Fig. 6A. To prove the specificity of mtDNA in this response, the authors should include (at least in one of these assays) liposomes containing pieces of chromosomal DNA as compared to liposomes with mtDNA.

Reviewer #3 (Comments to the Authors (Required)):

In this manuscript, Rabas et al investigate the role of metabotropic glutamate receptor mGluR3 in the regulation of Rab27-dependent EV (exosome) trafficking and secretion. This process is shown to require mitochondrial protein PINK-1 and mediates the packaging and secretion EV cargoes including mitochondrial proteins/DNA that promote tumor invasiveness in recipient cells in a TLR9 dependent manner. The authors also stimulate MDA-MB-213 breast cancer cells with artificial liposomes loaded with mtDNA and demonstrate this is sufficient to drive recycling of pro-invasive receptor MT1-MMP and $\alpha 5\beta 1$ integrin, from endosomes to the plasma membrane. Overall, this manuscript implicates tumor cell glutamate-regulated EV secretion as a driver of invasiveness and is consistent with the observed increase of circulatory glutamate in cancer patients and in mouse models of breast cancer. The authors demonstrate that EVs from donor cells can lead to pro-invasive signalling in recipient cells and that this can be recapitulated using liposomes loaded with mtDNA.

While the authors employ two methods to isolate EVs in their experiments (differential ultracentrifugation and sucrose density gradient centrifugation), downstream characterisation of these vesicles for specific exosome and non-exosome markers to rule out contaminate organelles/structures are lacking in several experiments. This is particularly concerning given the proposed role of mitochondrial proteins and nucleic acids in downstream functional assays where the presence of contaminating exosome-free mitochondria may potentially confound results. Further efforts to determine the purity and functional integrity of the isolated exosomes are needed to substantiate the claims proposed in this manuscript.

Specific comments:

- 1) CD63 is consistently used as a marker of exosomes in these preps throughout the paper. However, additional EV markers (e.g. Flotillin and CD9) are not shown after panel Figure 1a. These markers are needed for the interpretation of the data especially in panels where the reduction of CD63 is relatively minor (e.g. Figure 1c). Since Flotillin and CD9 were also shown to be reduced in EV preps following LY95 treatment (Fig 1a) and arguably to a higher degree than CD63, these markers should be assessed in the following panels in Figure 1 as well. Loading controls should also be consistently included for whole cell lysates in all panels.
- 2) The authors propose that PINK-dependent EVs are generated independently of autophagy/mitophagy or the newly describes LDELS pathway. However, the results do not support the authors' conclusion for a "complete lack" of involvement of ATG5/7 and FIP200 in mtDNA-containing exosome release. In Supplemental Figure 4, these experiments need to provide more rigorous and direct evidence that ATG 5 and 7 proteins are truly lost via western blot. Most importantly, the inhibition of autophagy as a result of ATG5/7 knockdown in Supp. Figure 4a is not at all convincing and nothing is provided on the degree of autophagy inhibition for FIP200 in Supp. Figure 4b. Additional evidence is required that autophagic flux is impaired upon FIP200 or ATG5/7 knockdown. Finally, LC3-II levels should be analysed in the EV preps to more convincingly rule out contributions for the ATG conjugation machinery.
- 3) The authors should consider the use of transmission electron microscopy as another measure of extracellular vesicle preparation purity and exclude the possibility contaminants.
- 4) In Figure 2, the authors claim mtDNA corresponding to genes encoding tRNA-Leu, COX1, ND2, ND6, CYB and 8339-9334 are more resistant to DNase digestion compared to control gene (β -

globin). However, only NADH-dehydrogenase 2 is assessed by qPCR in Figure 2d. Data for other genes listed is needed to robustly support the conclusion that mtDNA is broadly protected. Axis labels for Figure 2d-f should be changed from mtDNA to ND2 to more accurately reflect the quantification of a specific gene rather than "total mtDNA" as it currently appears.

5) Also, in Figure 2d, the degree of DNase "protection" between mtDNA (ND2) compared to β -globin is extremely modest. The authors should determine whether these relative differences between DNase treated vs. control samples are statistically significant between mtDNA (ND2) compared to β -globin and include details on statistical analysis in the figure legend. This experiment also appears to only have been performed once and error bars were derived from two technical replicates raising further questions about its robustness. Additional biological replicates are needed.

6) In Figure 2e and 3b, DNase protection assays should also be conducted in these experiments to corroborate that the inhibition of metabotropic glutamate receptors (LY95), CD63 and PINK specifically impacts the EV release of mtDNA. As discussed above, demonstration that multiple mitochondrial genes are protected is crucial to substantiate the claims that mtDNA being released within EVs.

7) In Figure 4, do EVs from siCD63 or siRab27 also lead to impaired MT1-MMP recruitment and gelatin degradation? It is important to determine whether treatment with CCCP or overexpression of PINK1 is able to rescue the impaired EV secretion following LY95 treatment.

8) In Figure 5 and 6, does TLR9 stimulation via ODN also enhance collagen degradation in MDA-MB-231 cells? Similarly, do siTLR9 recipient cells show impaired collagen degradation following stimulation with mtDNA exosomes or liposomes? It is also unclear from these experiments whether ODN-2006 added to cell culture medium and/or liposome packaged mtDNA are internalised via the same pathway to associate with TLR9-positive endosomes. This should be tested using fluorescently tagged ODNs or ethidium bromide stained mtDNA (if feasible) added extracellularly or loaded into liposomes and visualised via confocal microscopy.

Minor comments:

1) Although the EVs described here have many of the key characteristics associated with "exosomes", the term extracellular vesicle (EV) is now considered the preferred terminology in the field.

2) Figure 3 requires loading control for the cell lysates as well as other EV markers in addition to CD63.

3) In Supp. Figure 3c, does knockdown or knockout of PINK1 reduce the amount of Ub or phospho-Ub in EVs? This should be formally tested to more rigorously support the claim that PINK1 is important for the functional regulation of this process.

Reviewer #1

Major point 1: Although the decrease in mitochondrial proteins and DNA in exosomes following LY95 treatment is quite interesting, it is not expected that the absence of Cycophilin D protein would be greater than the absence of mtDNA (comparing Fig. 2b and 2e) in the exosomes. Also, it was already known that mitochondria are present in exosomes so the fewer exosomes in cells treated with the mGluR3 antagonist LY95 is around the same amount as the lower mitochondrial content (comparing Fig. 1a and Fig. 2f) is not unanticipated by the prior literature. The effect of LY95 on respiration may be very indirect (Fig. 2a).

Response:

We agree that, because mitochondrial components are known to be associated with extracellular vesicles (EVs), it is not unexpected that measures leading to inhibition of EV production – including inhibition of mGluR3 and knockdown of Rab27 – will reduce the release of both EV-associated mitochondrial DNA (mtDNA) and other mitochondrial components. However, because of the different methodologies that we have used to measure mitochondrial protein and mtDNA (Western blotting and qPCR respectively), we agree that it is unclear whether the release of EV-associated mitochondrial protein and mtDNA is influenced by the mGluR3 antagonist in a coordinate manner. We have, therefore, been more precise in wording statements describing these findings. For instance, we have modified the results text describing Fig. 2b and Figs. 2e so that it now reads: ‘inhibition of EV production - by addition of the mGluR3 antagonist - reduced the quantity of mitochondrial proteins associated with EV preparations (Fig. 2b)’; and ‘measures which lead to inhibition of EV production – such as inhibition of mGluR3 and knockdown of Rab27 or CD63 – lead to reduced release of mtDNA that was associated with EV preparations (Fig. 2e Supplementary Fig. 3b)’. Furthermore, we now show that overexpression of PINK1 increases release of CD63-positive EVs which are associated with mtDNA, and that a proportion of this mtDNA is protected from DNase and is, therefore, likely to be within the EVs (Fig. 4c–e).

To further strengthen data supporting the association of mtDNA with EVs, we have increased the range of primer sets used to measure components of the mitochondrial genome. Indeed, we have now repeated experiments in Fig. 2e using PCR primers hybridising with the mitochondrial COX1 gene, and these data are now shown in the new Fig. 2e (COX1 primers) and in Supplementary Fig. 3b (ND1 primers). Finally, we agree that the effect of LY95 on respiration is likely to be indirect. Our belief that this is the case is communicated by the results text describing Fig. 2a and Supplementary Fig. 2a-c. Indeed, this section is concluded by the statement: ‘These data indicate that proper function of the late endosomal/MVB network and EV production is required to maintain mitochondrial function.’, which we feel communicates

our feeling that the effect on mitochondrial respiration of inhibition of EV production (by LY95 addition, or knockdown of Rab27 or CD63) is not direct.

Major point 2: Figure 3 presents a main novel finding, that PINK1 is required for exosome release. However, the rationale for addressing this is not clear. As the authors note, PINK1 senses mitochondrial damage and an almost complete loss of mitochondrial membrane potential is required for PINK1 stabilization. The authors do show a minor loss of respiration but need to measure mitochondrial membrane potential using TMRM or TMRE. Does exposure of cells to glutamate as in Fig. 1b affect mitochondrial membrane potential? Rescue of the PINK1 siRNA is required in Fig. 3a regarding siPINK inhibition of exosome production and also for Fig. 3d.

Response:

As recommended by this reviewer, we have used TMRE to measure the consequences of inhibition of mGluR3 on the mitochondrial membrane potential. These data indicate that, although addition of CCCP leads to substantially reduced mitochondrial membrane potential (as expected), inhibition of mGluR3 for up to 5 days does not. These data are now included in Supplementary Fig. 2b. Moreover, although exposure of cells to glutamate (following glutamate starvation) leads to a small reduction in mitochondrial membrane potential, refeeding glutamate-starved cells with full medium does not. These data (shown below in **A**) indicate that the reduction in mitochondrial respiration which is apparent after 5 days of inhibition of mGluR3 (Fig. 2a) is not owing to loss of mitochondrial membrane potential.

We have now obtained rescue vectors for both wild-type and kinase-dead PINK1 using YFP-tagged PINK1s – kindly donated by Richard Youle's laboratory – into which we have introduced silent mutations to render them resistant to siRNA. This clearly indicates that expression of either wild-type or kinase-dead PINK1 completely rescues the reduction in EV release that we observe following PINK1 knockdown (Fig. 3c-d). However, incompatibility of

the fluorophores that we have used for the imaging in the old Fig. 3d with our rescue vectors has rendered it difficult to determine whether the effect of PINK1 knockdown on endosome-mitochondrion contact may be rescued by re-expression of PINK1.

Major point 3: The authors state that phospho-ubiquitin is highly abundant in exosomes based on Suppl. Fig. 3c. What they show is phospho-ubiquitin is detectable. How do the authors quantify phospho-ubiquitin to determine it is highly abundant? Relative to what? Total ubiquitin?

Response:

We agree with this comment and, as it is not possible to draw meaningful conclusions from Western blots re the abundance of phospho-ubiquitin, we have removed these data from the paper.

Major point 4: Using CCCP to test the effect of increased levels of PINK1 on exosome formation is very crude and susceptible to many off target effects of CCCP (such as Opa1 loss and thus mitochondrial fragmentation into smaller vesicles). Overexpression of PINK1 should be performed instead.

Response:

We have performed this experiment as recommended by this reviewer. These data indicate that overexpression of PINK1 increases release of CD63-positive EVs which are associated with mtDNA resistant to DNase. These data are now presented in Fig. 4c-e.

Major point 4: The authors nicely show that CCCP stabilizes PINK1 in Suppl. Fig. 2a. Western blots of PINK1 are also needed for the siRNA experiments (in addition to the nice phosphoUb experiments in Suppl. 2b). Most importantly, western blots of PINK1 are needed during exosome formation and when exosome production is inhibited by LY95.

Response:

We have now strengthened this aspect of the paper by performing Western blotting for PINK1 and cellular phospho-Ub content when PINK1 is stabilised by addition of CCCP, and when CCCP is added following the knockdown and re-expression either wild-type or kinase-dead PINK1. These data indicate clearly that knockdown of PINK1 opposes CCCP-driven increases in cellular phospho-Ub and that this is completely rescued by expression of wild-type, but not kinase-dead, PINK1 (Fig. 3c).

Major point 6: The authors indicate that PINK1 is required for mitochondrial contact with late endosomes (Fig. 3d). Does this contact require PINK1 enzyme activity? The

authors should compare mitochondrial contact with late endosomes after rescuing siPINK1 treated cells with kinase dead PINK1 relative to rescue with wild type PINK1. More mechanistic insight into how PINK1 mediates late endosome/multivesicular body contact is a key issue to address.

Response:

We agree that the mechanism by which PINK1-CD63 contacts mediate endosomal sorting of EVs is an important research question. However, we believe that addressing this is outwith the scope of the present study. Nevertheless, the experiments that we have performed in revising this paper indicate that rescue of either wild-type or kinase-dead PINK1 expression in PINK1 knockdown cells restore EV release to the same extent. This is even though kinase-dead PINK1 is, as expected, unable to restore CCCP-driven phospho-Ub levels in PINK1 knockdown cells (Figs. 3c, d). These observations indicate that the mechanism through which PINK1 drives EV release (and likely endosome-mitochondrial contact) is not dependent on PINK1's kinase activity. We have discussed this potentially non-canonical, kinase-independent mechanism through which PINK1 controls EV release in our discussion and look forward to addressing the details of this process in further studies.

Major point 7: The authors then show that "donor" exosomes influence the trafficking of MT1-MMP, a cell surface receptor involved in tumor invasion, and that inhibition of the DNA sensor, TLR9, or PINK1 inhibits trafficking of this marker of tumor cell invasiveness. To assess if mitochondrial DNA is required for tumor invasion, the authors use a very indirect approach. Liposomes containing mtDNA are incubated with MDA-MB-231 cells and shown to cause increased MT1-MMP trafficking to the ventral cell surface. This activity required TLR9 activity. Although interesting, this does not indicate mtDNA packaging into exosomes is required or involved in tumor invasiveness. A loss of mtDNA approach is needed. Producing Rho0 exosome donor cells and assessing activity of their exosomes would be one rigorous approach to test the conclusion.

Response:

To address this, we have generated mtDNA-deficient 'Rho0' cells by chronic treatment with ethidium bromine and collected EVs from these. Rho0 cells release the same quantity of EVs as do control cells. However, these EVs have substantially reduced mtDNA (but not nuclear DNA) content. Interestingly, EVs from Rho0 cells are unable to drive extension of invasive protrusions or to promote collagen degradation when they are incubated with glutamate-starved recipient tumour cells (Fig. 7a-d). These data indicate that mtDNA packaging into tumour EVs is involved in tumour invasiveness.

Minor point: At the end of page 6 the mention of fig 2a is incorrect.

This error has been corrected.

Reviewer #2

Major point 1: Figure 2D. The protection from DNAase of mtDNA as compared to beta-globin DNA seems not very convincing. There's only 30% residual mtDNA after treatment vs. 15% of beta-globin. Can the authors further comment and discuss this point? Is this difference biologically and statistically significant?

Response:

We have consolidated these data by conducting DNase treatment on an increased number of extracellular vesicle (EV) preparations (to allow statistical analysis) and by using an additional two PCR primer pairs to quantify both DNase-resistant and DNase-sensitive mtDNA (Fig. 2d). These data now indicate that approx. 10-15% of mtDNA (when assessed using primers hybridising with the COX1, ND1 or CYTB genes) in EV preparations is DNase-resistant. By contrast, only approx. 5% of β -globin DNA in the EV preparations is resistant to DNase. Statistical analysis (paired t-test) indicates that the difference between the proportion DNase resistant mtDNA (measured using any of the three aforementioned primer pairs) and β -globin DNA in the EVs is significant.

To address whether the presence of mtDNA and nuclear DNA in EV preparations is biologically significant, we generated mtDNA-deficient 'Rho0' cells by chronic treatment with ethidium bromine and collected EVs from these. Rho0 cells release the same quantity of EVs as do control cells, however, these EVs have a substantially reduced mtDNA (but not nuclear DNA) content. EVs from Rho0 cells are unable to drive extension of invasive protrusions or to promote collagen degradation when they are incubated with glutamate-starved recipient tumour cells (Fig. 7a-d). These data indicate that the mtDNA associated with EVs performed a biological role for which nuclear DNA cannot substitute.

Major point 2: Figure 2E. Please also provide Rab27 KD to confirm the proposed mechanism.

Response:

We have now determined that knockdown of Rab27 opposes release of EV-associated mtDNA to a similar extent as does inhibition of mGluR3 or knockdown of CD63 and these data are now included in a new Fig. 2e.

Major point 3: Figure 3D. The results shown here could be biased by the fact that CCCP treatment caused fragmentation of the mitochondria and this might affect the quantitation of contacts number. Can the authors further discuss this point in the text? In addition, can they also explain and further discuss why CCCP treatment does not increase the number of contacts, but only their duration?

Response:

We acknowledge that CCCP may generate off-target effects, and interpretation of results from experiments involving this reagent will be complicated by these. To control for off-target effects of CCCP, we have increased PINK1 levels by overexpression of YFP-tagged PINK1 and shown that this increases release of CD63-positive EVs containing mtDNA, which is resistant to DNase (Fig. 4c-e). However, due to incompatibility of the fluorophores that we use to image CD63 and PINK1 (GFP and YFP, respectively), it has proved technically very challenging to perform the experiments to determine the effect of overexpression of PINK1 on endosome-mitochondrial contacts. We are unable to explain why CCCP does not increase the number of contacts, but only their duration. However, knockdown of PINK1 clearly suppresses the number of contacts established between endosomes and mitochondria both in the presence and absence of CCCP.

Major point 4: Figure 4C. Please provide the effect of RAB 27 KD on gelatin degradation assay.

Response:

We have now performed gelatin-degradation assays on glutamine-starved recipient cells that have been treated with EVs from control and Rab27 knockdown donor cells. This indicates that EVs from Rab27 knockdown donor cells have significantly reduced ability to drive gelatin degradation (Fig. 6c). This is consistent with a role for Rab27 in release of mtDNA-containing exosomes which drive invasiveness in recipient cells.

Major point 5: Figure 5D-E. To prove the in trans cell-to-cell communication, authors should perform the TLR9 KD in the recipient cell providing exosomes produced by control donor cells (as shown in the scheme in panel A).

Response:

We have performed TLR9 knockdown in recipient MDA-MB-231 cells and tested their ability to respond to EVs produced by control donor cells. This indicates that TLR9 knockdown cells are unable to respond to EVs from control donor cells by mounting an increased invasive response – as assessed both by extension of invasive protrusions and by the collagen degradation assay (Fig. 8c-d).

Major point 6: Fig. 6A. To prove the specificity of mtDNA in this response, the authors should include (at least in one of these assays) liposomes containing pieces of chromosomal DNA as compared to liposomes with mtDNA.

Response:

To demonstrate the specificity of mtDNA in the invasive response of recipient cells to EVs, we have generated Rho0 cells which are depleted of mtDNA. As stated above, EVs from Rho0 cells have reduced mtDNA (but not nuclear DNA) content and are unable to drive extension of invasive protrusions or to promote collagen degradation when they are incubated with glutamate-starved recipient tumour cells (Fig. 7a-d). These data prove the specificity of mtDNA over nuclear DNA in evoking invasive responses in recipient cells.

Reviewer #3

Major point 1: CD63 is consistently used as a marker of exosomes in these preps throughout the paper. However, additional EV markers (e.g. Flotillin and CD9) are not shown after panel Figure 1a. These markers are needed for the interpretation of the data especially in panels where the reduction of CD63 is relatively minor (e.g. Figure 1c). Since Flotillin and CD9 were also shown to be reduced in EV preps following LY95 treatment (Fig 1a) and arguably to a higher degree than CD63, these markers should be assessed in the following panels in Figure 1 as well. Loading controls should also be consistently included for whole cell lysates in all panels.

Response:

We have performed additional Western blots and repeated these experiments where necessary. We now include flotillin and CD9 Western blot analyses of extracellular vesicle (EV) preparations from cells incubated in the presence/absence of glutamate and following knockdown of CD63 and Rab27 (Fig. 1a, b and Supplementary Fig. 1a, b). Moreover, we have now included loading controls for the whole-cell lysates of all these experiments. We have also provided CD9 Western blots of the EV preparations, and cell lysate loading controls for the PINK1 knockdown experiments (Fig. 3a).

Major point 2: The authors propose that PINK-dependent EVs are generated independently of autophagy/mitophagy or the newly describes LDELS pathway. However, the results do not support the authors' conclusion for a "complete lack" of involvement of ATG5/7 and FIP200 in mtDNA-containing exosome release. In Supplemental Figure 4, these experiments need to provide more rigorous and direct evidence that ATG 5 and 7 proteins are truly lost via western blot. Most importantly, the inhibition of autophagy as a result of ATG5/7 knockdown in Supp. Figure 4a is not at all convincing and nothing is provided on the degree of autophagy inhibition for FIP200 in Supp. Figure 4b. Additional evidence is required that autophagic flux is impaired upon FIP200 or ATG5/7 knockdown. Finally, LC3-II levels should be analysed in the EV preps to more convincingly rule out contributions for the ATG conjugation machinery.

Response:

We have performed Western blots to illustrate the efficiency of the ATG5/7 knockdown (Supplementary Fig. 5a). Moreover, we have now included Western blots to demonstrate that knockdown of FIP200 reduces starvation-induced cleavage of LC3B (Supplementary Fig. 5c). Finally, we have measured the levels of LC3B in EVs (see below in **B**). These data indicate that LC3B-II is certainly present in EVs, as reported by Leidal et al (Nature Cell Biology 22:187). Moreover, as one would expect, knockdown of ATG5/7 or FIP200 leads to reduced

levels of LC3B-II in EVs. However, knockdown of ATG5/7 or FIP200 does not reduce the quantity of EVs released (Supplementary Fig. 5b, d).

Thus, we concur with this reviewer that these data do not provide evidence for a ‘complete lack’ of involvement of ATG5/7 and FIP200 in EV release. Rather, they indicate that the release of CD63-positive, mtDNA-containing EVs is strongly dependent on PINK1, but less so on ATG5/7 and FIP200. Moreover, our new evidence that EV release is restored in PINK1 knockdown cells by expression of kinase-dead PINK1 (despite that this construct is unable to rescue phospho-Ub levels) indicates that PINK1’s contribution to EV release is via a mechanism distinct from manner which this protein contributes to mitophagy. We have now moderated and refined our discussion to accommodate the nuances that these new data introduce to the interpretation of our data.

Major point 3: The authors should consider the use of transmission electron microscopy as another measure of extracellular vesicle preparation purity and exclude the possibility contaminants.

Response:

As recommended by this reviewer, we have now used immuno-gold labelling in combination with transmission electron microscopy (TEM) to identify CD63-positive EVs released from MDA-MB-231 cells. These data indicate that the morphology of structures present in our EV preparations conform to that of exosomes/EVs vesicles, and that they encompass a size-distribution of 60–200 nm with a median size of approx. 90 nm. Importantly, this analysis indicates clearly that inhibition of mGluR3 (using LY95) leads to a significant reduction in the quantity of CD63-positive EVs released by MDA-MB-231 cells (Fig. 1c).

Major point 4: In Figure 2, the authors claim mtDNA corresponding to genes encoding tRNA-Leu, COX1, ND2, ND6, CYB and 8339-9334 are more resistant to DNase digestion compared to control gene (β -globin). However, only NADH-dehydrogenase 2 is assessed by qPCR in Figure 2d. Data for other genes listed is needed to robustly support the conclusion that mtDNA is broadly protected. Axis labels for Figure 2d-f should be changed from mtDNA to ND2 to more accurately reflect the quantification of a specific gene rather than "total mtDNA" as it currently appears.

Response:

To increase the robustness of our data supporting the conclusion that mtDNA is broadly protected from DNase, we have repeated several DNase protection assays and assessed mtDNA using primers recognising three mitochondrial genes: COX1, CYTB and ND1. These experiments have now been performed on 3 independent occasions and, from this, we have found that the protection of DNA encoding the COX1, CYTB ND1 genes is significantly greater than that of the (nuclear) β -globin gene (paired t-test) (Fig. 2d). Furthermore, as recommended by this reviewer, throughout the paper, we have now labelled the graphs presenting qPCR data to indicate the specific mitochondrial gene being quantified.

Major point 5: Also, in Figure 2d, the degree of DNase "protection" between mtDNA (ND2) compared to β -globin is extremely modest. The authors should determine whether these relative differences between DNase treated vs. control samples are statistically significant between mtDNA (ND2) compared to β -globin and include details on statistical analysis in the figure legend. This experiment also appears to only have been performed once and error bars were derived from two technical replicates raising further questions about its robustness. Additional biological replicates are needed.

Response:

Please see our response to this reviewer's previous comment (Major point 5).

Major point 6: In Figure 2e and 3b, DNase protection assays should also be conducted in these experiments to corroborate that the inhibition of metabotropic glutamate receptors (LY95), CD63 and PINK specifically impacts the EV release of mtDNA. As discussed above, demonstration that multiple mitochondrial genes are protected is crucial to substantiate the claims that mtDNA being released within EVs.

Response:

In part, this comment is addressed by our response to this reviewer's major point 5, and we have also included qPCR analysis for an additional mitochondrial gene (COX1) when analysing EVs released by LY95-treated and CD63 and Rab27 knockdown cells (Fig. 2e).

Furthermore, to address the specific role of mtDNA (as opposed to nuclear DNA) in the pro-invasive function of EVs we have generated Rho0 cells which have reduced mtDNA content. Rho0 cells release similar quantities of EVs as control MDA-MB-231 cells, and these EVs are associated with nuclear DNA, but very little detectable mtDNA (Fig. 7b). Interestingly, although they are associated with some nuclear DNA, these mtDNA-depleted EVs are ineffective in evoking invasive responses in recipient cells (Fig. 7c, d). These data indicate that mtDNA (but not nuclear DNA) associated with EVs has a particularly important role to play in evoking TLR-mediated responses.

Major point 7: In Figure 4, do EVs from siCD63 or siRab27 also lead to impaired MT1-MMP recruitment and gelatin degradation? It is important to determine whether treatment with CCCP or overexpression of PINK1 is able to rescue the impaired EV secretion following LY95 treatment.

Response:

As recommended by this reviewer, we have now conducted gelatin-degradation assays using EVs from control and Rab27 knockdown cells. These data indicate that EVs from Rab27 knockdown cells have significantly reduced ability to promote gelatin degradation by glutamate-starved recipient cells (Fig. 6c). We have also now overexpressed PINK1 (using an expression vector encoding PINK1-YFP) and shown that this is sufficient to drive release of EVs associated with mtDNA (Fig. 4c-e).

Major point 8: In Figure 5 and 6, does TLR9 stimulation via ODN also enhance collagen degradation in MDA-MB-231 cells? Similarly, do siTLR9 recipient cells show impaired collagen degradation following stimulation with mtDNA exosomes or liposomes? It is also unclear from these experiments whether ODN-2006 added to cell culture medium and/or liposome packaged mtDNA are internalised via the same pathway to associate with TLR9-positive endosomes. This should be tested using fluorescently tagged ODNs or ethidium bromide stained mtDNA (if feasible) added extracellularly or loaded into liposomes and visualised via confocal microscopy.

Response:

We have performed additional experiments to show that siRNA of TLR9 in glutamate-starved recipient cells reduces their ability to degrade gelatin in response to EVs from donor cells (Fig. 8c, d). Furthermore, we have now shown that pharmacological activation of TLR9 (using the TLR agonist, ODN) increases the ability of glutamine-starved MDA-MB-231 cells to degrade gelatin (Fig. 8a). Due to constraints of time and resources (including limited access to the laboratory during the COVID19 lockdown restrictions), we have not yet investigated the mechanism through which mtDNA EVs are internalised by recipient cells. We anticipate that

this will be an exciting area of investigation and will look forward to conducting these studies in the future.

Minor comment 1: Although the EVs described here have many of the key characteristics associated with "exosomes", the term extracellular vesicle (EV) is now considered the preferred terminology in the field.

Response:

We agree and, to conform with the current preferred nomenclature, we have changed our term of reference from exosomes to extracellular vesicles (EVs) throughout the paper.

Minor comment 2: Figure 3 requires loading control for the cell lysates as well as other EV markers in addition to CD63.

Response:

We have now included these data (Fig. 3a)

Minor point 3: In Supp. Figure 3c, does knockdown or knockout of PINK1 reduce the amount of Ub or phospho-Ub in EVs? This should be formally tested to more rigorously support the claim that PINK1 is important for the functional regulation of this process.

Response:

In the series of experiments that were aimed at establishing whether overexpression of PINK1 can rescue EV release following siRNA of PINK1, it became clear to us that both wild-type (kinase-active) and kinase-dead PINK1s have similar capacity to promote EV release (Fig. 3d). Moreover, this is even though kinase-dead PINK1 is not able to restore cellular phospho-Ub levels following PINK1 knockdown (Fig. 3c). This indicates that the mechanism through which PINK1 controls EV release is likely to be independent from its ability to phosphorylate ubiquitin. We have discussed the evidence for a non-canonical (kinase-independent) role for PINK1 in EV release in our discussion. However, we feel that further studies addressing the details of how PINK1 controls EV release in a kinase-independent manner lie outwith the scope of the present study.

August 26, 2021

RE: JCB Manuscript #202006049R

Prof. Jim C Norman
CRUK-Beatson Institute
Integrin Cell Biology Lab Beatson Institute for Cancer Research Garscube Estate, Switchback Road
Glasgow, East Dumbartonshire G61 1BD

Dear Jim,

Thank you for submitting your revised manuscript entitled "PINK1 drives production of mtDNA-containing extracellular vesicles to promote invasiveness". The original three reviewers were markedly split in their recommendations, with one highly expert reviewer recommending rejection due to the fact the revisions have brought up new important issues regarding the mechanism by which Pink1 acts. The other two reviewers recommend acceptance, with one reviewer ranking your study of particularly high priority for publication. We agree with the negative reviewer that the very surprising finding that EV production is rescued quite effectively by even kinase-dead Pink1 could lead to an intriguing project. However, the original editorial decision letter stated, "Although exploring further the mechanisms of PINK1 action in late endosome/MVB contact would indeed be quite intriguing, it is not in our minds as essential as providing data supporting the current specific conclusions." Therefore, we find that requiring a mechanistic explanation of the novel, intriguing findings would be somewhat like moving the goalposts and not consistent with the practices of JCB. Although the mechanistic basis is now regrettably unclear, we hope that this will be resolved by a future in-depth follow up study. Therefore, as recommended by the two positive reviewers we agree to publish your paper in JCB pending final revisions necessary to meet our formatting guidelines (see details below).

A. MANUSCRIPT ORGANIZATION AND FORMATTING:

Full guidelines are available on our Instructions for Authors page, <https://jcb.rupress.org/submission-guidelines#revised>. **Submission of a paper that does not conform to JCB guidelines will delay the acceptance of your manuscript.**

- 1) Text limits: Character count for Articles is < 40,000, not including spaces. Count includes title page, abstract, introduction, results, discussion, acknowledgments, and figure legends. Count does not include materials and methods, references, tables, or supplemental legends.
- 2) Figures limits: Articles may have up to 10 main text figures.
- 3) Figure formatting: Scale bars must be present on all microscopy images, * including inset magnifications. Molecular weight or nucleic acid size markers must be included on all gel electrophoresis.
- 4) Statistical analysis: Error bars on graphic representations of numerical data must be clearly

described in the figure legend. The number of independent data points (n) represented in a graph must be indicated in the legend. Statistical methods should be explained in full in the materials and methods. For figures presenting pooled data the statistical measure should be defined in the figure legends. Please also be sure to indicate the statistical tests used in each of your experiments (either in the figure legend itself or in a separate methods section) as well as the parameters of the test (for example, if you ran a t-test, please indicate if it was one- or two-sided, etc.). Also, if you used parametric tests, please indicate if the data distribution was tested for normality (and if so, how). If not, you must state something to the effect that "Data distribution was assumed to be normal but this was not formally tested."

5) Abstract and title: The abstract should be no longer than 160 words and should communicate the significance of the paper for a general audience. The title should be less than 100 characters including spaces. Make the title concise but accessible to a general readership.

6) Materials and methods: * Should be comprehensive and not simply reference a previous publication for details on how an experiment was performed. Please provide full descriptions in the text for readers who may not have access to referenced manuscripts. *

7) Please be sure to provide the sequences for all of your primers/oligos and RNAi constructs in the materials and methods. * You must also indicate in the methods the source, species, and catalog numbers (where appropriate) for all of your antibodies. Please also indicate the acquisition and quantification methods for immunoblotting/western blots. *

8) Microscope image acquisition: The following information must be provided about the acquisition and processing of images:

- a. Make and model of microscope
- b. Type, magnification, and numerical aperture of the objective lenses
- c. Temperature
- d. Imaging medium
- e. Fluorochromes
- f. Camera make and model
- g. Acquisition software
- h. Any software used for image processing subsequent to data acquisition. Please include details and types of operations involved (e.g., type of deconvolution, 3D reconstitutions, surface or volume rendering, gamma adjustments, etc.).

10) * Supplemental materials: There are strict limits on the allowable amount of supplemental data. Articles may have up to 5 supplemental figures. Please therefore reduce your SI figure count and be sure to correct the callouts in the text to reflect this change. Please also note that tables, like figures, should be provided as individual, editable files. A summary of all supplemental material should appear at the end of the Materials and methods section.

13) ORCID IDs: ORCID IDs are unique identifiers allowing researchers to create a record of their various scholarly contributions in a single place. At resubmission of your final files, please consider providing an ORCID ID for as many contributing authors as possible.

B. FINAL FILES:

Thank you for this interesting contribution, we look forward to publishing your paper in Journal of Cell Biology.

Sincerely,

Kenneth Yamada, MD, PhD
Editor

Andrea L. Marat, PhD
Senior Scientific Editor

Journal of Cell Biology

Reviewer #1 (Comments to the Authors (Required)):

I will summarize my review of the revised manuscript as - one step forward, two steps backward. In response to the reviewers' comments the authors deleted the data on phospho-ubiquitin (Pink1 kinase product) localization in extracellular vesicles, which is logical as phospho-ubiquitin no longer seemed related to Pink1 mechanism. This stems from the authors new experiments rescuing the EV defect in Pink1 knock down cells with both WT and Kinase Dead Pink1. Surprisingly, KD Pink1 rescued EV production as well as WT. Since Pink1 induces EV formation without kinase activity, the authors are at the beginning of an interesting project - that is to decipher how Pink1 does this. Pink1 has a mitochondrial targeting motif, a membrane spanning domain and a kinase domain, making it straightforward to interrogate Pink1 mechanism beyond kinase activity. Without such mechanistic insights I feel this is not well suited for JCB.

Reviewer #2 (Comments to the Authors (Required)):

The authors have addressed all the major concerns that I have raised in the first round of revision. Results presented are relevant, carefully controlled and suitable for publication.

Reviewer #3 (Comments to the Authors (Required)):

I have reviewed the revised manuscript and the authors' response to my critiques of the previous version. My concerns have been largely addressed with new experimental data.